# Tumor cell villages define the co-dependency of tumor and microenvironment in liver cancer

Meng Liu [1], Maria O. Hernandez[2], Darko Castven[3], Hsin-Pei Lee[1], Wenqi Wu[1], Limin Wang[4], Marshonna Forgues [4], Jonathan M. Hernandez[5], Jens U. Marquardt [3] ✉ & Lichun Ma [1,6] ✉

Spatial cellular context is crucial in shaping intratumor heterogeneity. However, understanding how each tumor establishes its unique spatial landscape and what factors drive the landscape for tumor fitness remains significantly challenging. Here, we analyze over 2 million cells from 50 tumor biospecimens using spatial single-cell imaging and single-cell RNA sequencing. We develop a deep learning-based strategy to spatially map tumor cell states and their surrounding environmental architecture, and find that different tumor cell states can be organized into distinct clusters, or "villages," each supported by unique microenvironments. Notably, tumor cell villages exhibit village-specific molecular co-dependencies between tumor cells and their microenvironment and are associated with patient outcomes. Perturbation of molecular co-dependencies via random spatial shuffling of the microenvironment results in destabilization of the corresponding villages. We validate our findings using single-cell, spatial, and bulk transcriptome data from 740 liver cancer patients. This study provides insights into understanding tumor spatial landscape and its impact on tumor aggressiveness.

Intratumor heterogeneity (ITH) is a major hurdle for effective interventions across a broad range of cancers. It has been increasingly studied by single-cell transcriptome analysis, where each tumor may contain various transcriptomic states distinguished by unique gene expression patterns[1–3]. In pan-cancer single-cell transcriptome studies, diverse tumor cell transcriptomic states have been defined, including cell cycle, epithelial-to-mesenchymal transition (EMT), and major histocompatibility complex (MHC) II-related states, among others[4,5]. While these features of ITH are well observed, the mechanisms by which diverse tumor cells spatially coordinate and how each tumor establishes its unique spatial landscape remain poorly understood. Spatial transcriptomic profiling of various cancers at the spot resolution (a mixture of cells in each spot) has begun to unravel the intratumor spatial landscape, where a non-random spatial distribution of diverse transcriptome-based tumor clusters is observed within individual tumors[6–13]. In ovarian cancer, a specific tumor cell state was found to spatially colocalize with tumor-infiltrating lymphocytes based on single-cell spatial transcriptome analysis[14]. These observations suggest that the spatial cellular context is crucial in shaping ITH and further motivated us to understand the spatial landscape of individual tumors.

A critical challenge is how to spatially define tumor cell landscapes and identify the drivers underlying diverse spatial architectures. Tumors from different patients evolve independently, leading to

[1]Cancer Data Science Laboratory, Center for Cancer Research, National Cancer Institute, Bethesda, MD, USA. [2]Spatial Imaging Technology Resource, Center for Cancer Research, National Cancer Institute, Bethesda, MD, USA. [3]Department of Medicine I, University Medical Center, Lübeck, Germany. [4]Laboratory of Human Carcinogenesis, Center for Cancer Research, National Cancer Institute, Bethesda, MD, USA. [5]Surgical Oncology Program, Center for Cancer Research, National Cancer Institute, Bethesda, MD, USA. [6]Liver Cancer Program, Center for Cancer Research, National Cancer Institute, Bethesda, MD, USA. ✉e-mail: Jens.Marquardt@uksh.de; lichun.ma@nih.gov

unique gene expression profiles that reflect patient-specific characteristics. Directly clustering spatially resolved tumor spots or cells within each tumor tends to capture patient-specific patterns, making it difficult to compare the landscapes across patients. Here, we introduce the concept of "villages" to define the distinct landscapes formed by tumor cells, where each village arises from the spatial coordination of diverse tumor cell transcriptomic states supported by specific local environments, termed the Spatial Dynamics Network (SDN), surrounding individual tumor cells. Tumor cell villages represent the smallest community within a tumor. This concept draws inspiration from human villages, which are small, close-knit communities where individuals share resources and responsibilities. Like human villages, tumor cell "villages" may use communal mechanisms to promote growth and enhance defense, reducing individual cell vulnerability. The introduction of this village concept may provide insights into how tumor cells establish distinct spatial landscapes within individual patients, while also enabling comparisons of these landscapes across patients. More importantly, the defined tumor cell villages may uncover unique molecular co-dependencies among cells that are crucial for orchestrating their organization within each village, offering opportunities for therapeutic interventions.

In this study, we focus on primary liver cancer (PLC) due to its well-known, extensive ITH, making it an ideal model to investigate the spatial relationships of diverse tumor cell populations within individual tumors[1,6,15–18]. We perform transcriptomic profiling of liver tumors at single-cell spatial resolution using CosMx™ SMI 1000-plex in situ spatial imaging platform[19]. Additionally, we conduct single-cell RNA-sequencing (scRNA-seq) on tumors from the same patients to generate more comprehensive expression profiles of individual cells. In total, we profile ~2.4 million cells from 50 tumor biospecimens of 7 patients with PLC, including 4 with hepatocellular carcinoma (HCC) and 3 with intrahepatic cholangiocarcinoma (iCCA). We define tumor cell transcriptomic states and the SDNs surrounding individual tumor cells. We observe distinct spatial preferences of tumor cell states and the coordination of diverse tumor cells as tumor cell "villages" supported by unique SDNs. Tumor cell villages are found to be linked to patient outcomes. Notably, village-specific molecular co-dependencies between tumor cells and their microenvironment are observed, the perturbation of which leads to the destabilization of the corresponding tumor cell villages. We validate our findings using single-cell transcriptome data, 10× Visium spatial transcriptome data, and bulk transcriptome data from 766 samples of 740 PLC patients. This study may provide crucial insights into understanding diverse tumor spatial architectures and their impact on tumor aggressiveness.

## Results

### Single-cell spatial characterization of tumors from primary liver cancer patients

To comprehensively capture ITH and investigate its stability within individual tumors, we designed the study to collect samples from multiple locations within each tumor from PLC patients (Fig. 1A). In total, 50 samples were collected from the tumor core, tumor border, and adjacent non-tumor tissue of 7 PLC patients (4 HCC, 3 iCCA; Supplementary Data 1). Among them, 15 samples (whole tissue section) were applied for single-cell spatial transcriptome profiling using the CosMx™ SMI 1000-plex in situ imaging platform. The remaining samples were profiled by single-cell RNA-sequencing (scRNA-seq) in our previous study using the 10× Genomics droplet-based 5′ assay[20]. In this study, we refer to the CosMx™ SMI data as "single-cell spatial data" and the scRNA-seq data as "single-cell data". After quality control (Methods), spatial transcriptomic profiles of 2,347,589 cells were derived (Fig. 1B, Supplementary Fig. 1A, B). In addition, scRNA-seq data of 112,506 cells were generated (Fig. 1B)[20]. When mapping individual cells to their spatial locations, we observed a strong concordance among cell type annotations, the expression of cell type-specific

marker genes, and protein staining, demonstrating a good representation of the tumor landscape with the single-cell spatial data (Fig. 1C–E, Supplementary Fig. 1B, C; Supplementary Data 2).

To further determine the quality of the single-cell spatial transcriptome data, we compared it with the scRNA-seq data from the same patients. We found strong correlations in gene expression between the two platforms, confirming a high quality of the spatial data (Supplementary Fig. 1D). Interestingly, we observed a striking difference in cell type composition from the two platforms. In the scRNA-seq data, T cells accounted for over 80% of the entire cell population (Fig. 1F). By contrast, epithelial cells represented the major cell type in the single-cell spatial data, which was consistent with the histological images of the tumors[20]. This discrepancy may be due to tissue dissociation, storage, and library preparation processes in scRNA-seq, during which immune cells may be preferentially preserved[21]. These observations suggest that the CosMx platform may offer an unbiased cellular map of liver tumors.

We distinguished malignant cells from non-malignant epithelial cells in the single-cell spatial data based on their transcriptomes and geographic locations in tumors (Supplementary Fig. 1E). Specifically, we performed clustering analysis of all epithelial cells and mapped each cluster to its spatial location. Clusters that mapped to tumor regions were labeled as malignant cells, whereas those mapped to adjacent non-tumor regions were defined as epithelial cells based on histological images (see Methods). We did not infer copy number variations from the transcriptome to confirm the malignancy of epithelial cells due to a limited number of targeted genes in the single-cell spatial data. In total, 792,192 malignant cells and 1,555,397 non-malignant cells were identified from the single-cell spatial data. Malignant and non-malignant cells from the scRNA-seq data have been determined from our previous analysis[20].

### Heterogeneous transcriptomic states in malignant cells

To understand the degree of ITH in PLC, we determined the transcriptomic states across all malignant cells from the single-cell spatial data using non-negative matrix factorization (NMF), a method has been successfully applied in defining tumor cell states based on scRNA-seq data[4,5]. We used scRNA-seq data as a reference to increase confidence in cellular states identification (Methods). Overall, we identified 12 distinct tumor cell states across all malignant cells tested, including 9 unique states of cell cycle, stress response, immune response and locomotion, metallothionein, EMT, hepatocyte-like, cholangiocyte-like, MHC-II, interferon response, as well as 3 mixed states of stress/metallothionein, stress/immune response and locomotion, and MHC-II/metallothionein (Fig. 2A, B, Supplementary Fig. 2A–C). These states were annotated based on the most highly expressed gene modules and their functional enrichment (Fig. 2B; Supplementary Data 3). Immune response and locomotion (CCL20, CXCL5, ICAM1) represented the most prevalent cellular states in malignant cells, highlighting the interactions between tumor cells and immune cells (Supplementary Fig. 2D). Other frequent states included stress/metallothionein, cholangiocyte-like, and cell cycle (Supplementary Fig. 2D). Interestingly, we observed a small proportion of MHC II-related tumor cells. Although MHC II molecules are primarily expressed by professional antigen-presenting cells, tumor cells can also express these molecules, and their specific expression by tumor cells has been linked to favorable outcomes in cancer patients[22]. In contrast to malignant cells, non-malignant epithelial cells were mainly enriched in metallothionein and stress-associated programs, which are related to the functions of hepatocytes (Supplementary Fig. 2D). We found most of the states were similar to those identified in pan-cancer studies, suggesting commonalities of tumor cell transcriptomic states among cancer types[4,5]. However, hepatocyte- and cholangiocyte-like states were uniquely observed in liver cancer, reflecting organ-specific cell lineages. Noticeably, each tumor contained a mixture of different

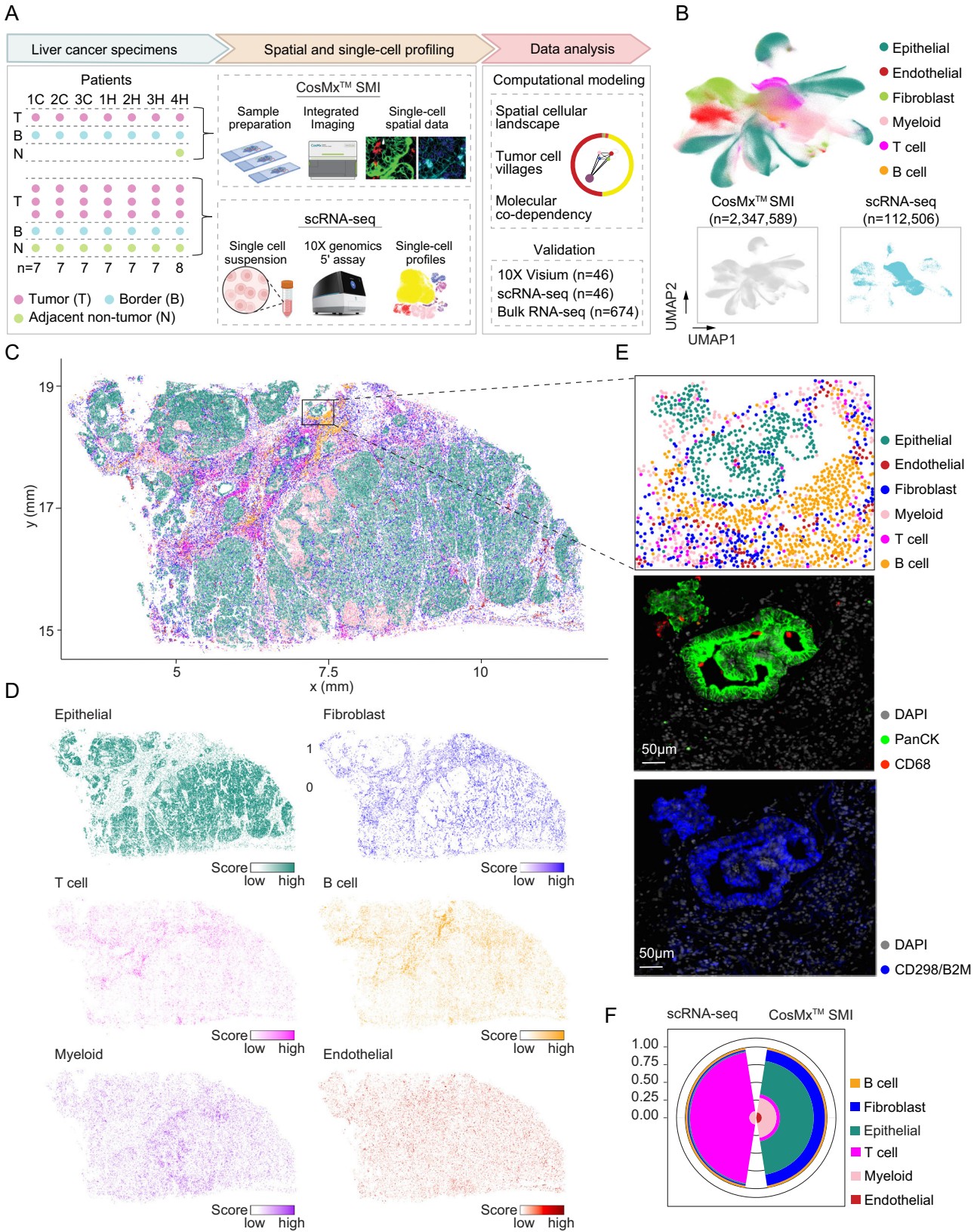

tumor cell states, underscoring pronounced intratumor heterogeneity within each lesion (Supplementary Fig. 2E; Supplementary Data 4).

## Landscape of non-malignant cells

We delineated the landscape of non-malignant cells by performing Harmony-based clustering of each cell type using the single-cell spatial transcriptome data[23]. In total, we identified 79 clusters, comprising 18 clusters of endothelial cells, 14 clusters of fibroblasts, 16 clusters of myeloid cells, 10 clusters of B cells, and 21 clusters of T cells, reflecting a complex and heterogeneous landscape of immune and stromal cells in liver cancer (Fig. 2C and Supplementary Fig. 3). We further annotated these clusters into 44 cellular states based on differentially

**Fig. 1 | Single-cell spatial transcriptomic profiling of primary liver cancer.**
**A** Schematic overview of the workflow in this study. A total of 50 samples collected from the tumor core (T), tumor border (B), and adjacent non-tumor tissue (N) of 7 liver cancer patients (4 HCC, 3 iCCA) were profiled. Sample IDs were named based on histological subtypes of liver cancer, where H represents HCC and C represents iCCA. Single-cell transcriptome data, 10X Visium spatial transcriptome data, and bulk transcriptome data were used for validation, with sample numbers indicated. Illustration was partially created using BioRender. **B** UMAP embeddings of all profiled single cells colored by cell types ($n = 2,460,095$, top panel), and the cells colored by profiling methods ($n = 2,347,589$, CosMx™ SMI; $n = 112,506$, scRNA-seq,

bottom panel). **C**, **D** A representative tumor sample (1CT) colored by cell types (**C**) and gene score of each cell type (**D**). Gene score was determined based on the average expression of marker genes specific to each cell type. **E** Cell type annotation and protein staining of a selected window in (**C**). CD68 (red) and Pan-cytokeratin (Pan-CK, green) represent markers for macrophages and epithelial cells. DAPI (light gray) and CD298/B2M (blue) were used for nuclei and membrane staining. A total of 15 samples were profiled using CosMx™ SMI. Scale bars, 50 μm. **F** Comparison of cell type compositions based on scRNA-seq and CosMx™ SMI from the same set of liver cancer patients in our cohort.

expressed genes (Fig. 2C and Supplementary Fig. 3). For example, in the fibroblast population, matrix fibroblasts (*COL1A1*), antigen-presenting fibroblasts (*CD74*, *IGKC*), smooth muscle cells (*RGS5*), EMT-like fibroblasts (*KRT17*), tumor-like fibroblasts (*VEGFA*), *CXCL12*+ CAFs (*CXCL12*, *IGF2*), *IL32*+ CAFs (*IL32*, *CXCL10*), and *CCL19*+ CAFs (*CCL19*) were determined. Similarly, different cellular states in endothelial cells, myeloid cells, and lymphoid cells were defined (Fig. 2C). We validated cluster robustness by comparing them with clusters derived from individual patients and found that most patient-derived clusters mapped predominantly to specific Harmony-based clusters, indicating minimal over-correction by Harmony (Supplementary Fig. 4). To determine if the non-malignant cell subtypes resolved from the single-cell spatial data and the scRNA-seq data were consistent, we compared the clusters in each major cell type derived from the two approaches. We found that most clusters from the spatial data could be well-matched to specific clusters in the scRNA-seq data, indicating that the spatial approach effectively captures major cellular information despite targeting fewer genes than scRNA-seq (Supplementary Fig. 5). Interestingly, we observed some unique clusters in the spatial data. For example, in myeloid cells, the *TUBB*+ proliferative tumor-associated macrophages (TAMs) were only found in the spatial data, suggesting that the single-cell spatial method may identify potential novel cell clusters (Supplementary Fig. 5)[24,25]. Collectively, the single-cell spatial transcriptomic profiles resolved by CosMx offer a detailed landscape of non-malignant cells in liver cancer.

**Spatial distribution of cells within and across tumor regions**
To assess intratumor heterogeneity, we analyzed the spatial distribution of cell (sub)types within and across tumor regions. Overall, epithelial cells and T cells were markedly decreased in both the tumor border and core relative to adjacent non-tumor tissue, whereas myeloid cells and fibroblasts gradually increased from non-tumor region to the border and core (Supplementary Fig. 6A). Among epithelial cells, malignant cells increased and non-malignant epithelial cells decreased from the non-tumor region toward the tumor core (Supplementary Fig. 6A). Detailed comparison of tumor cell states revealed largely comparable proportions between the border and core regions in our cohort (Supplementary Fig. 6B). While most non-malignant cell subtypes showed no significant differences across these regions, several displayed region-specific enrichment. At the tumor border, two immune subtypes (Kupffer cells and *CD8*+ effector T cells) and four stromal subtypes (liver sinusoidal endothelial cells [LSECs], post-capillary venule endothelial cells, *CCL19*+ CAFs, and *CXCL12*+ CAFs) were significantly enriched compared to the tumor core (Supplementary Fig. 6C). This observation was consistent with the typical localization of Kupffer cells and LSECs in normal liver tissue. In contrast, tip cells and smooth muscle cells were elevated in the tumor core, along with reg-TAMs, suggesting a potential pro-angiogenic and immunosuppressive microenvironment (Supplementary Fig. 6C)[26–29].

We also investigated the spatial variation of cells within each tumor region by calculating Moran's I to quantify spatial autocorrelation. For all major cell types, Moran's I values were positive, indicating that each cell type exhibited some degree of spatial autocorrelation within individual tumor regions (Supplementary Fig. 7A). Notably,

Moran's I increased for epithelial cells, myeloid cells, fibroblasts, and endothelial cells in the tumor border and core compared with adjacent non-tumor regions, suggesting stronger spatial clustering of these populations within tumors (Supplementary Fig. 7A). In contrast, Moran's I values for T cells and B cells decreased in the tumor border and core relative to non-tumor regions, reflecting a more dispersed distribution within the tumor (Supplementary Fig. 7A). These observations are consistent with the clustered distribution of T cells and B cells around portal veins in normal liver, whereas other cell types are more evenly dispersed in normal tissue[30]. Malignant cell states displayed low but positive Moran's I values, with no significant differences between border and core, suggesting only marginal spatial autocorrelation of each malignant cell state within individual tumor regions for the samples in our cohort (Supplementary Fig. 7B).

**Characterization of the surroundings of individual malignant cells by defining spatial dynamics network**
A successfully evolved tumor represents a spatially well-organized ecosystem, where malignant cells actively interact with their local environment. This dynamic interplay continuously shapes tumor cell functions, fueling ITH. To uncover the roles of spatial context in driving ITH, it's essential to delineate the surroundings of each individual malignant cell. The resolved single-cell spatial data in this study provide a unique and powerful approach for this purpose. We characterized the surroundings of each malignant cell by determining the composition of its neighbors within 40 μm, as in our previous study (Fig. 3A)[31]. We further performed clustering analysis of all malignant cells based on their surroundings using the Louvain algorithm (Fig. 3A). Consequently, malignant cells falling into the same cluster shared similar spatial neighbors. This strategy is independent of transcriptome-based clustering, where tumor cells within a certain cluster have similar transcriptomes. We tested the stability of the malignant cell clusters by applying different radii of 20 μm, 60 μm, 80 μm, and 100 μm, where similar results were found to those derived from 40 μm, demonstrating stable clusters of malignant cells using the neighborhood information (Supplementary Fig. 8A, B). In total, 14 clusters of malignant cells (corresponding to 14 types of surrounding scenarios) with distinct surrounding environments were determined (Fig. 3B, C, Supplementary Fig. 8C, D). We annotated different types of surrounding environments based on their compositions and designated each type as a Spatial Dynamics Network (SDN) around malignant cells (Fig. 3A, C). Vascularized tumors represented a major group of SDNs (SDN-T1–SDN-T5), with endothelial cells in the surroundings of malignant cells. We also identified a tumor-dominant SDN (SDN-T8), in which malignant cells comprised the majority of surrounding cells. Additionally, we found five SDNs (SDN-T10–SDN-T14) with an enrichment of myeloid cells and fibroblasts, as exemplified in an iCCA sample (Fig. 3C, D). These findings highlight the diverse microenvironments surrounding malignant cells in liver cancer.

**Tumor transcriptomic states are related to their spatial contexts**
To understand the roles of spatial context in shaping ITH, we analyzed the relationship between tumor cell transcriptomic states and the SDNs. Specifically, we performed an enrichment analysis of malignant

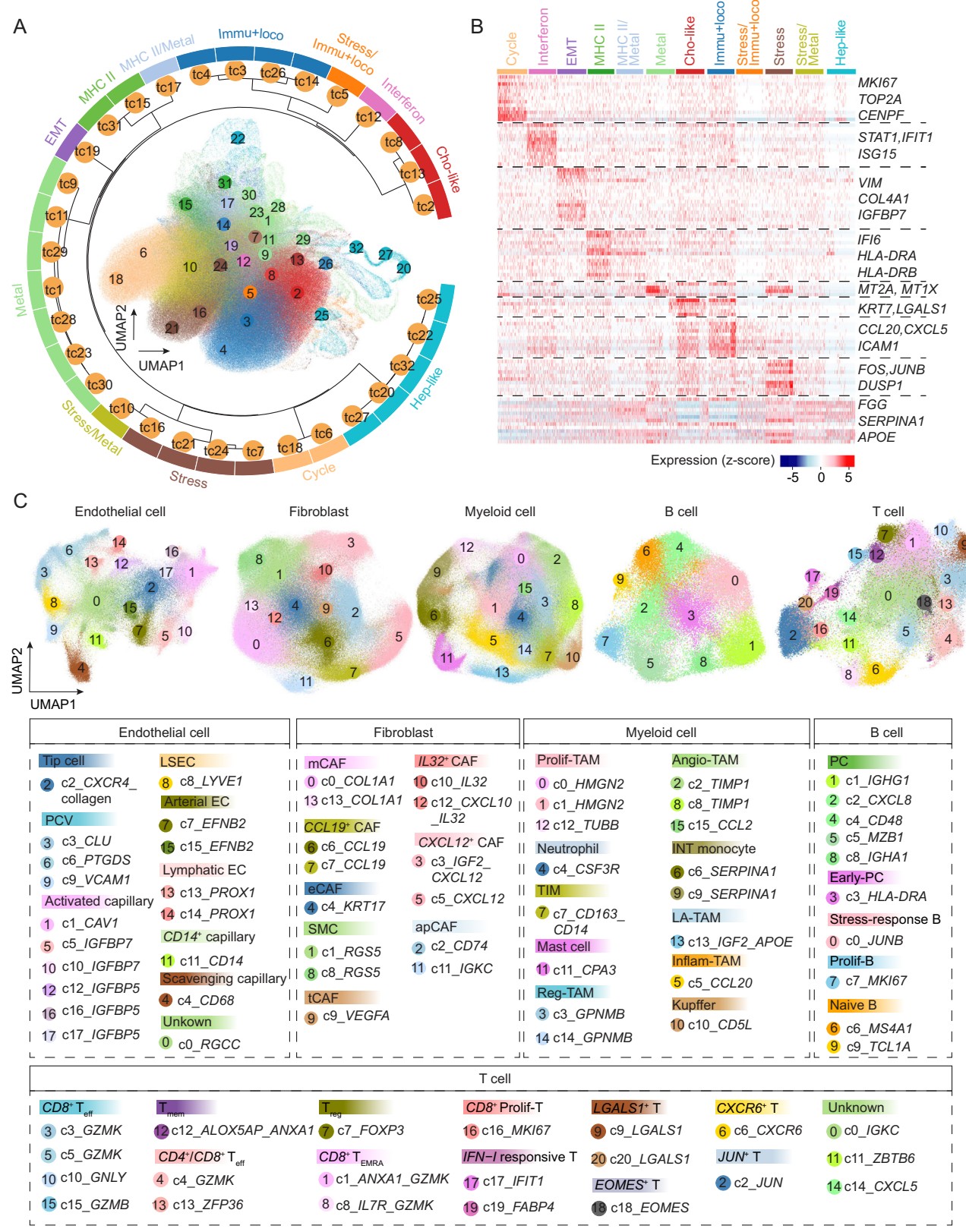

cell states on the SDNs using a random shuffling strategy (Method). Notably, tumor cell states displayed non-random spatial distributions, with each state preferentially enriched in specific SDNs (Fig. 4A). For example, EMT-like tumor cells were enriched in SDNs (SDN-T1–SDN-T5) that were related to vascularized tumors. Consistently, an increased density of endothelial cells along with EMT transition has been reported[32]. In addition, we found EMT-like tumor cells were in

significantly closer spatial proximity to endothelial cells compared to the tumor cell states that were enriched in completely different SDNs (Fig. 4B). We observed unique populations of IGF2+ lipid-associated macrophages (LA-TAM) in SDN-T1 and SDN-T2, where EMT was the most strongly enriched (Figs. 3C and 4A). IGF2 has been demonstrated to be secreted by macrophages to regulate EMT[33]. We also found CXCL12+ CAFs expressing IGF2 and CXCL12 in SDN-T1 and SDN-T2,

**Fig. 2 | Landscape of cells in liver cancer. A** Transcriptomic state of malignant cells. UMAP of malignant cells colored by clusters (inner plot) and a radial dendrogram indicating the hierarchical relationship of the clusters (outer dendrogram). Clusters were determined based on gene module scores of malignant cells (see Methods for details). Cho-like cholangiocyte-like, EMT epithelial-to-mesenchymal transition, Hep-like hepatocyte-like, Immu+loco immune response and locomotion, Metal metallothionein. **B** The gene expression profiles related to each tumor cell transcriptomic state in (**A**). Representative genes were indicated. **C** UMAP of non-malignant cells. Clusters of each non-malignant cell type were determined and annotated based on marker genes. EC endothelial cell, PCV postcapillary venule, LSEC liver sinusoidal endothelial cell, CAF cancer-associated fibroblast, mCAF matrix CAF, eCAF EMT-like CAF, SMC smooth muscle cell, tCAF tumor-like CAF, apCAF antigen-presenting CAF, TAM tumor-associated macrophage, Prolif-TAM proliferating TAM, TIM tumor infiltrating monocyte, Reg-TAM immune-regulatory TAM, Angio-TAM pro-angiogenic TAM, INT monocyte intermediate monocyte, LA-TAM lipid-associated TAM, Inflam-TAM inflammatory cytokine-enriched TAM, PC plasma cell, Prolif-B proliferative B cell, $T_{eff}$ effector T cell, $T_{mem}$ Memory T cell, $T_{EMRA}$ recently activated effector memory or effector T cell, Treg regulatory T cell, Prolif-T proliferative T cell.

suggesting a pivotal role of the local environment in regulating EMT phenotype of tumor cells (Fig. 3C). In contrast to the complex local microenvironment of EMT-like tumor cells, cell cycle-related tumor cells were strongly enriched in tumor-dominant SDNs (SDN-T7–SDN-T9) with few stromal and immune cells. This was further supported by cell density analysis, with a higher density of tumor cells surrounding cell cycle-related tumor cells than others (Fig. 4C). Additionally, we noticed that cell cycle- and EMT-related tumor cells were enriched in completely different SDNs. It's well known that tumor cells going through EMT have a decreased ability to proliferate[34]. We further demonstrated distinct local environments of the two tumor cell states. Interferons are cytokines that help trigger the immune system to eradicate pathogens or tumor cells[35]. We observed an enrichment of interferon response-related tumor cells in SDN-T10 and SDN-T2, where myeloid cells and *CD8*+ T cells were found. These tumor cells expressed markers of *IFIT1* and *ISG20*, which are related to antiviral defenses[36,37]. Noticeably, interferon response-related tumor cells were mainly found in a patient with infections of the hepatitis C virus and human immunodeficiency virus in our cohort, suggesting a potential antiviral response in this patient (Supplementary Fig. 2E; Supplementary Data 1). We also found a strong enrichment of cholangiocyte-like tumor cells and immune response and locomotion-related tumor cells in the SDNs with fibroblasts and myeloid cells (SDN-T11–SDN-T13). Ligand-receptor interaction analysis between tumor cell states and their local environments demonstrated enriched communication patterns in each state, further supporting the spatial preference of different tumor cell states (Supplementary Fig. 9). For example, interactions mediated by VEGF were mainly observed between EMT-related tumor cells and their surrounding environments (Supplementary Fig. 9).

To validate the spatial preference of tumor cell states, we analyzed liver tumor samples from three publicly available datasets based on the 10× Visium platform[6,38,39]. Since 10× Visium data is at a spot (55 μm in diameter) resolution, the methods designed for single-cell spatial data cannot be directly applied to determine cellular states and SDNs here. Therefore, we developed pseudo-bulk gene signatures for each tumor cell transcriptomic state and SDN by averaging the associated genes from our single-cell spatial data (Methods). We further applied these gene signatures to the three datasets and used a correlation-based approach to determine the enrichment of each tumor cell state in the SDNs (Methods). We observed highly consistent enrichment patterns with those found in the single-cell spatial data, with a much stronger enrichment of tumor cell states in SDNs that showed significant association in the single-cell spatial analysis compared to other SDNs (Fig. 4D). Collectively, these observations suggest that the transcriptomic states of tumor cells are not randomly distributed within a tumor. Instead, they are closed associated with their surrounding microenvironments, underlying the potential roles of spatial context in driving ITH.

### Tumor cell villages are identified using graph attention networks

Diverse tumor cells may spatially coordinate to collectively drive tumor growth and progression, conferring survival advantages to the tumor and enhancing its resilience against interventions[40]. We define this spatial coordination of diverse tumor cell transcriptomic states, supported by specific local environments, as tumor cell "villages." Like human villages, tumor cell "villages" may use communal mechanisms to promote growth and enhance defense, reducing individual cell vulnerability.

To determine tumor cell villages, we used graph attention networks, which are deep learning neural networks specifically designed for analyzing graph-structured data[41]. These networks leverage an attention mechanism to enhance feature learning by weighing the importance of neighboring nodes, allowing us to capture the complex spatial relationships between diverse tumor cell states. In general, malignant cells within a tumor were represented as connected graphs, where nodes indicated individual cells characterized by their cellular states and SDNs, and edges linked nodes within a 40 μm distance, consistent with the SDN analysis (Fig. 5A). The embeddings derived from the graph attention networks were then applied for clustering analysis to define tumor cell villages (Fig. 5B). We excluded three small clusters with fewer than 5000 cells each, accounting for less than 1% of total malignant cells. Eight distinct tumor cell villages were then identified (Fig. 5C and Supplementary Fig. 10A). Village 1 and Village 2 contained a mix of tumor cell states with a vascularized environment. By contrast, Village 4 and Village 7 were dominated by cell cycle-related tumor cells, akin to cancer germinal centers. Villages 3, 5, and 6 were enriched in cholangiocyte-like and immune response and locomotion-related tumor cells, surrounded by a microenvironment of fibroblasts and macrophages, reflecting a well-developed ecosystem of tumor cells with ample resources. Village 8 was primarily associated with immune response and locomotion. As an example, 1CB (an iCCA sample) featured Villages 3, 5, and 6 in distinct geographic locations (Fig. 5D). We observed that tumor cell village compositions were more consistent across regions within the same patient than across different patients in our cohort (Supplementary Fig. 10B).

We determined surrogate markers for tumor cell villages through differential gene expression analysis (Supplementary Fig. 10C; Supplementary Data 5). The identified genes were found largely specific to each tumor cell village and can accurately predict the presence of their corresponding villages (Supplementary Fig. 10D–F), suggesting that these genes may effectively serve as surrogates for defining tumor cell villages. Using these genes, we validated the identified tumor cell villages with spatial transcriptome data from 46 HCC or iCCA tumors profiled on the 10X Visium platform[6,38,39,42] (Supplementary Fig. 11A, B). To examine how tumor cell villages relate to patient prognosis, we performed survival analysis on 37 HCC or iCCA patients with single-cell transcriptome data[1], stratifying the patients by surrogate gene-based hierarchical clustering. We found that patients enriched for Villages 3–8 gene signatures had significantly poorer overall survival compared to those with features related to Villages 1 and 2 (Fig. 5E and Supplementary Fig. 11C). A consistent trend was observed when restricting the analysis to HCC patients alone (Supplementary Fig. 11D). We didn't analyze iCCA patients separately, as all were predominantly enriched in Villages 3–8. The survival associations were further confirmed using bulk transcriptomic data from 674 HCC patients in the TCGA, LCI (Liver Cancer Institute)[43], and Mongolia[44] cohorts, where consistent

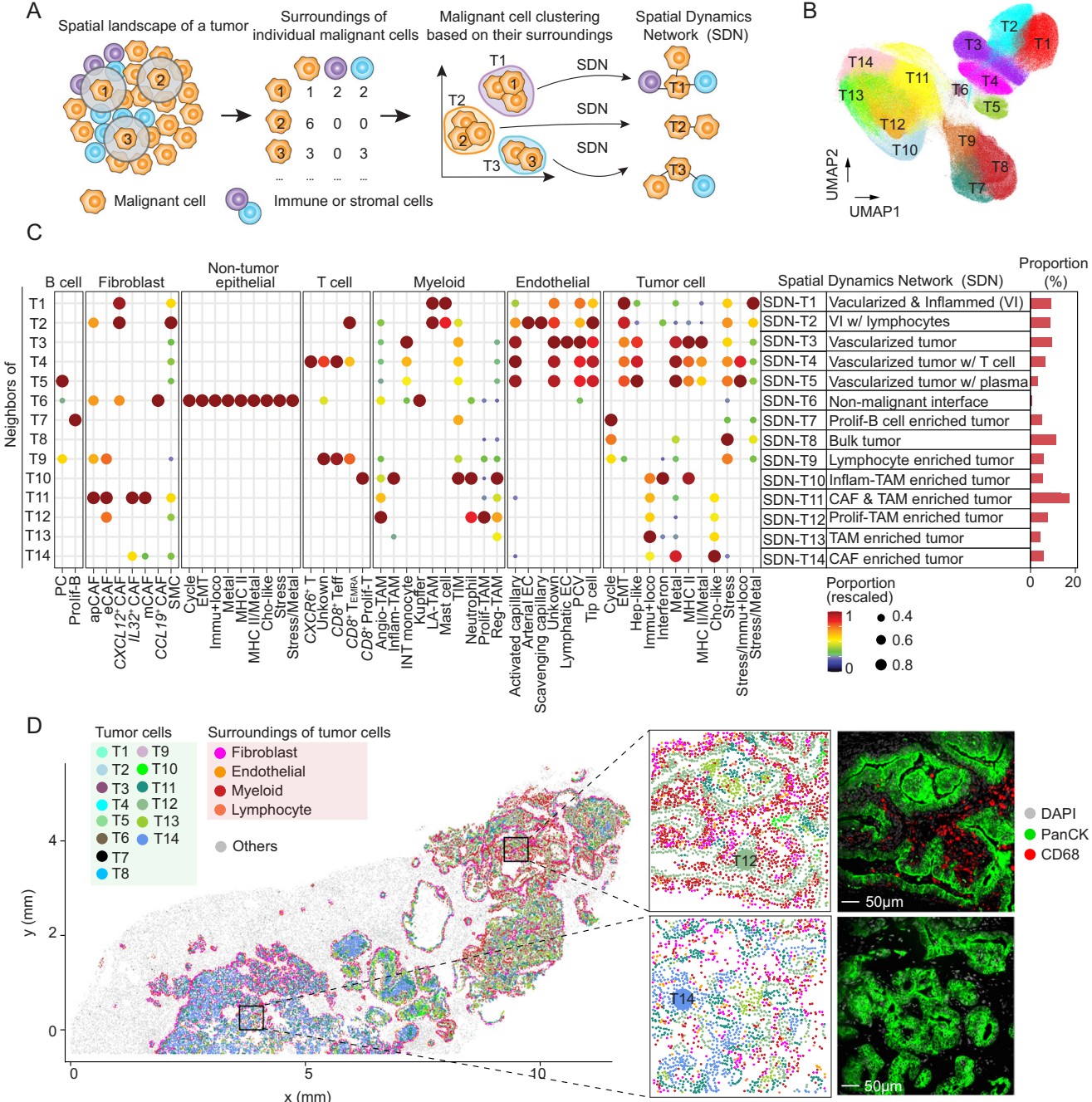

**Fig. 3 | The SDNs surrounding individual malignant cells. A** Schematic overview of SDN determination. For a targeted malignant cell, its surroundings were determined based on a 40 μm distance. Malignant cells were further characterized by the compositions of their surroundings and were subjected to clustering analysis. The average surrounding composition of each malignant cell cluster was used to define SDNs. **B** Malignant cell clusters determined based on the approach in (**A**). Malignant cells within the same cluster share similar surrounding features. See also Supplementary Fig. 8C. **C** Dot plot of the surrounding compositions for each malignant cell cluster in (**B**). Each row of the dot plot represents the composition of the surroundings of a malignant cell cluster. The color and size of the dots indicate the scaled (scaled to [0, 1] by column) proportion of cell state/subtype. Only dots with scaled values ≥ 0.2 are displayed in this figure to present the major enriched cell state/subtypes. See also Supplementary Fig. 8D. **D** A representative example (1CB, left panel) of malignant cells colored by their SDNs. Malignant cells were indicated using cool colors, while the surrounding non-malignant cells were shown in warm colors. Other cells are shown in gray. Two representative windows were selected with corresponding protein staining. A total of 15 samples were profiled using CosMx™ SMI. Scale bars, 50 μm.

links between tumor cell villages and clinical outcomes were observed (Fig. 5F–H and Supplementary Fig. 11E). These analyses highlight the presence of well-coordinated tumor cell villages in liver cancer and suggest that tumor cell villages are linked to patient outcomes.

To investigate the mechanisms underlying the observed clinical associations, we performed gene set enrichment analysis. Villages 1 and 2 were enriched in metabolic-related pathways, such as bile acid metabolism, adipogenesis, and fatty acid metabolism (Supplementary Fig. 11F). In contrast, pathways such as MYC targets, E2F targets, G2/M checkpoint, mitotic spindle, reactive oxygen species pathway, PI3K/AKT/mTOR signaling, and angiogenesis were observed across multiple Villages within 3–8, corresponding to programs that promote cell

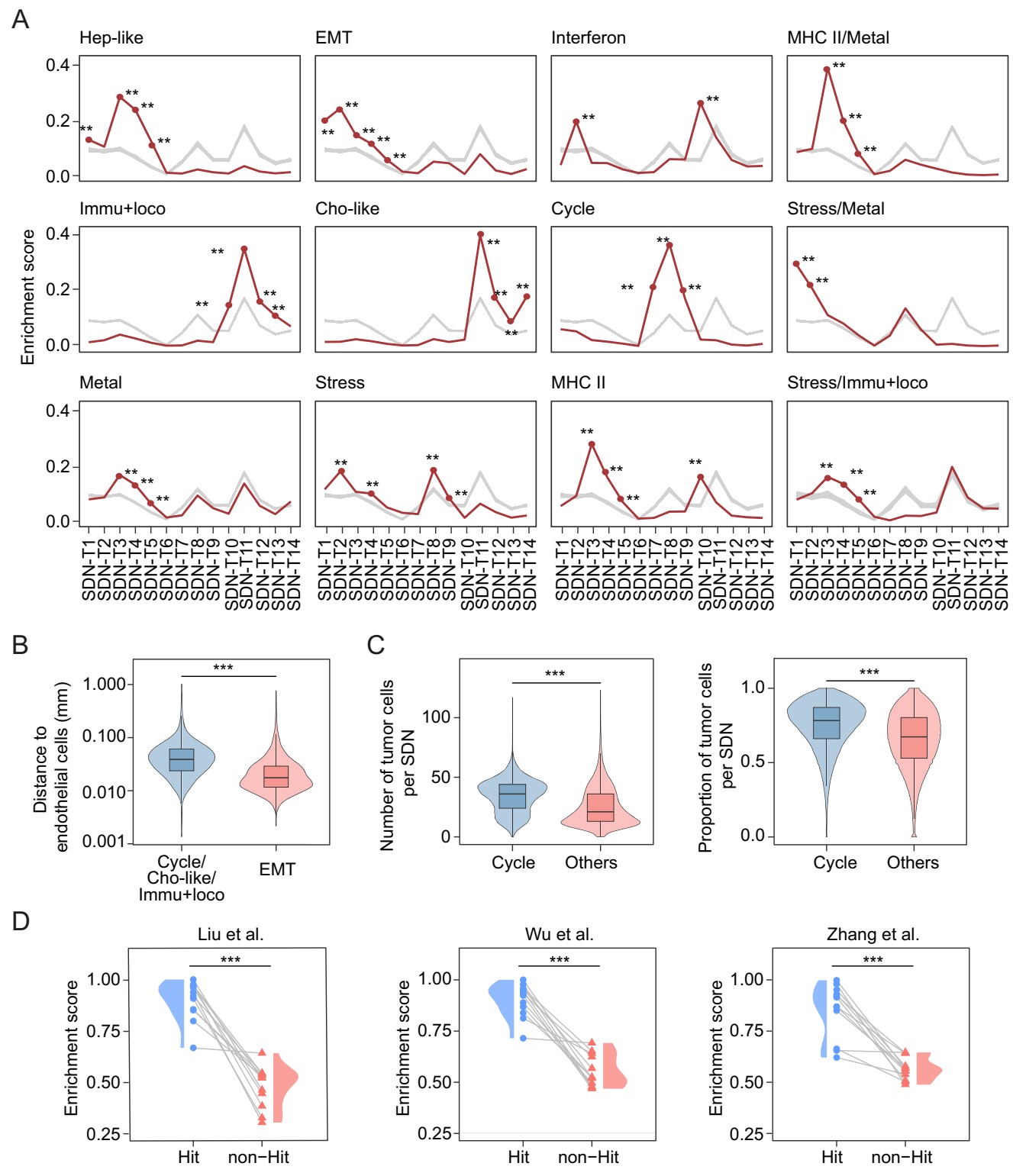

growth, survival, resistance to cell death, and neovascularization (Supplementary Fig. 11F). Although a small subset of EMT-like cells was detected in Village 2, EMT pathways were not enriched. Notably, while Villages 1 and 2 contained more endothelial cells, the endothelial cells in Villages 3–8 exhibited significantly higher expression of tumor-specific endothelial signatures, consistent with the angiogenesis pathway observed in several of these villages (Supplementary Fig. 11G)[45]. We did not observe significant differences in the overall abundance of inflammatory-related fibroblasts (*CXCL12*+ CAFs, *IL32*+ CAFs, and *CCL19*+ CAFs) between Villages 1–2 and Villages 3–8

(Supplementary Fig. 11H). Collectively, these results suggest that Villages 3–8 may represent more aggressive tumor ecosystems compared with Villages 1 and 2, which may underlie their unfavorable clinical outcomes.

## Molecular co-dependencies linked to tumor cell villages

To uncover key factors driving cell coordination within each tumor cell village, we developed a method to evaluate the molecular co-dependencies between tumor cells and non-tumor (stromal or immune) cells. Specifically, we calculated the correlations between

**Fig. 4 | Enrichment of tumor cell states on SDNs. A** Enrichment of each tumor cell state on SDNs. Red lines indicate observed enrichment while gray lines stand for randomized enrichment score. *P*-values were calculated based on the observed and randomized values (*n* = 100 iterations per condition, see Methods for details). No adjustment was made for multiple comparisons. \**, *p*-value < 0.01. Source data and exact *p*-values are provided as a Source data file. **B** The distances between EMT-like malignant cells and endothelial cells (*n* = 20,312), as well as the distances between other malignant cells (cell cycle, cholangiocyte-like, or immune response and locomotion) and endothelial cells (*n* = 433,456). Each box shows the median (center line), interquartile range (box), and data range (whiskers). Only tumor cell states with no significant enrichment in the SDN-T1–SDN-T5 were included in the comparison with EMT-like malignant cells. *P*-value was calculated with a one-sided Student's *t*-test. \*\*\*, *p*-value < 0.001. Detailed statistics and source data are provided as a Source data file. **C** The number (left) and the proportion (right) of malignant cells in the surrounding (within 40 μm distance) of cell cycle-related malignant cells (*n* = 125,867) compared to that of other malignant cells (*n* = 665,674). Each box shows the median (center line), interquartile range (box), and data range (whiskers). *P*-value was calculated using one-sided Student's *t*-test. \*\*\*, *p*-value < 0.001. Detailed statistics and source data are provided as a Source Data file. **D** Validation of the spatial preference of tumor cell states using 10X Visium data from Liu et al. (left), Wu et al. (middle), and Zhang et al. (right). Each pair of connected dot and triangle represents a tumor cell state. For each tumor cell state, "Hit" represents the significantly associated SDNs in (A), while "non-Hit" indicates the rest of the SDNs (*n* = 12 tumor cell states). Violin plots of the distributions of the enrichment scores were shown. *p*-value was calculated with a one-sided paired Student's *t*-test. \*\*\*, *p*-value < 0.001. Source data and detailed statistics are provided as a Source Data file.

each pair of genes from tumor and non-tumor cells based on a 40 μm distance, with an assumption that highly correlated gene pairs are strongly co-dependent (Methods). Using this method, we identified the top correlated gene pairs from each village and found that these gene pairs are village-specific, with significantly higher correlation scores in their respective villages compared to others (Fig. 6A; Supplementary Data 6). We further evaluated the gene-pair correlations in each village by increasing the spatial distance thresholds from 40 μm to 80 μm, 160 μm, 320 μm, and by extending beyond the village boundaries. Notably, a gradual decline in correlation scores was observed as the spatial separation of cells increased, underscoring the spatial specificity of the gene-pair relationships (Fig. 6B). To demonstrate the importance of these gene-pair relationships to each tumor cell village, we employed a random forest model to predict village identities of the tumor cells based on the identified gene pairs. We observed a significant decline in prediction accuracy when the non-tumor cells surrounding the tumor cells within each village were randomly shuffled, compared to the accuracy achieved with the original village configurations (Fig. 6C). These findings suggest that molecular co-dependencies may be crucial for maintaining the integrity of each village, and their perturbation may potentially destabilize or even collapse the village structure.

Among the top-ranked molecules from tumor cells, we focused on serine protease inhibitor Kazal-type 1 (*SPINK1*) due to its unique expression in tumor cells (Fig. 6D and Supplementary Fig. 12A). *SPINK1*-related gene pairs were predominantly found in Villages 6, with a few detectable in Village 3 (Fig. 6E). Most of the *SPINK1*-associated genes were matrix CAFs (mCAFs)-related genes, including *COL1A1*, *COL1A2*, *COL5A1*, *COL4A1*, *BGN*, and *MGP* (Fig. 6F; Supplementary Data 6). This spatial relationship was validated using Visium transcriptome data, which showed that *SPINK1*+ spots were significantly closer to the paired gene-enriched spots than expected by random chance (Fig. 6G). Multiplex immunofluorescence for SPINK1 and COL1A1 further confirmed the colocalization of the two proteins in patient 3C, who had the highest proportion of Village 6 among all patients (Supplementary Figs. 10B and 12B). Gene set enrichment analysis suggested that *SPINK1*+ tumor cells were associated with metastasis-related features, while *SPINK1*- tumor cells were enriched in metabolism- and coagulation-related pathways (Fig. 6H). Additionally, single-cell transcriptome analysis demonstrated that *SPINK1*+ tumor cells engaged more extensively with mCAFs than other CAFs through ligand-receptor interactions (Fig. 6I; Supplementary Data 7 and 8). *SPINK1* itself was identified as one ligand in these interactions with *EGFR* serving as its receptor (Supplementary Data 7)[46,47]. Consistently, *SPINK1* downstream genes were expressed at significantly higher levels in mCAFs compared with other CAFs (Supplementary Fig. 12C). Similarly, the *EGFR* downstream genes were significantly upregulated in mCAFs (Supplementary Fig. 12D). When the relationship of *SPINK1* and its paired genes were perturbed by randomly shuffling non-tumor cells, we found that the prediction accuracy of village identities by the

random forest model declined significantly compared with the original village configurations, with a greater decrease observed in Village 6 than in Village 3, consistent with the higher enrichment of *SPINK1*-related gene pairs in Village 6 (Fig. 6J). The prediction accuracy was further declined when all the identified correlated genes in these villages were perturbed (Fig. 6J). These observations suggest the molecular co-dependencies between *SPINK1*+ tumor cells and mCAFs may play an important role in sustaining the integrity of Villages 3 and 6. Collectively, these findings indicate that the defined tumor cell villages may reveal unique molecular co-dependencies among cells with each village, opens avenues for developing treatment strategies aimed at disrupting these co-dependencies to destabilize the tumor.

## Discussion

Over the past decade, transcriptomic ITH has been increasingly recognized, owing to the development of single-cell technologies. However, an essential factor in understanding ITH—the spatial context of cells—is lost in single-cell analysis. To determine the geographic landscape of diverse tumor cell transcriptomic states and their spatial relationships, we performed single-cell spatial transcriptomic profiling of liver cancer. We developed a bioinformatics method to classify the surroundings of individual malignant cells, where 14 different types of SDNs were identified, including vascularized tumor, tumor-dominant environment, and tumor-nontumor interface, among others. Interestingly, we found that each tumor cell state was preferentially associated with specific SDNs, rather than being randomly distributed throughout the tumor. The observations were further supported by spatial proximity analysis, cell density analysis, and ligand-receptor interaction analysis. These findings indicate a close link between tumor cell states and their microenvironments, suggesting a crucial role of spatial context in driving ITH. Notably, such insights would not be possible without spatial information of individual cells, highlighting the importance of single-cell spatial transcriptome approaches in advancing our understanding of ITH.

Tumor cells form diverse spatial landscapes for survival fitness. Understanding how each tumor establishes its unique spatial landscapes and what factors drive these landscapes may offer new insights into disease development, progression, as well as response to therapy. To this end, we developed a graph attention neural network-based method to identify spatially coordinated groups of tumor cells, which we termed tumor cell "villages." This concept was inspired by human villages, which historically formed to fulfill social, economic, and environmental needs, fostering a sense of community, mutual support, security, and the ability to collectively adapt to environmental challenges. Analogously, tumor cell villages center on malignant cells, representing functional communities of diverse tumor cell states that are spatially organized and supported by specific microenvironments. Within these villages, tumor cells cooperate for growth and survival, thereby reducing individual cell vulnerability. This concept differs from tumor microenvironment, which describes the broader

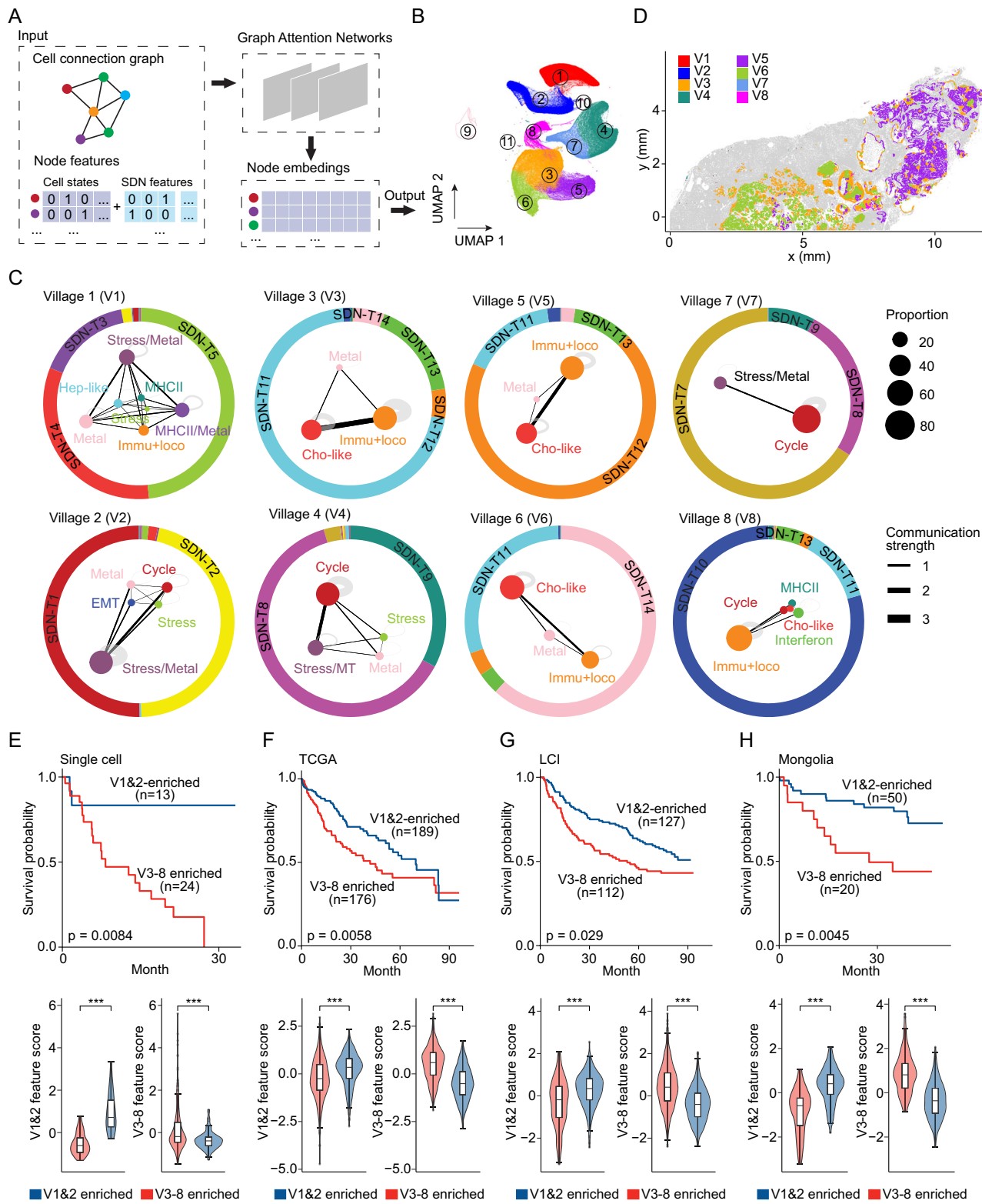

ecosystem of non-malignant components surrounding malignant cells, and from tumor niche, which refers to microanatomical structures that maintain specific tumor cells, such as cancer stem cells[48].

PLC is one of the most lethal malignancies worldwide, with a median overall survival of 6–10 months and a 5-year survival rate of around 20%[49,50]. HCC and iCCA are the two major clinical subtypes of PLC. Extensive ITH in both clinical subtypes has been demonstrated using single-cell approaches in our previous studies[1,15,16,51]. Thus, in this study, we used PLC to model the spatial relationships among diverse

malignant cells. We identified eight different types of villages among available liver tumors using graph attention neural networks. We observed that most patients harbored a combination of different tumor cell villages and that these villages can be detected across patients. Given the limited number of patients in the discovery cohort, we further examined village enrichment using a larger single-cell transcriptome dataset that included both HCC and iCCA patients[1]. We found that Villages 1 and 2 were preferentially enriched in HCC, whereas Villages 3–8 were enriched in both HCC and iCCA, as

**Fig. 5 | Tumor cell villages in liver cancer. A** Schematic overview of determining tumor cell villages using a graph attention network-based approach. Malignant cells were represented as connected graphs, where nodes indicate individual cells characterized by their cellular states and SDNs, and edges link nodes within a 40 μm distance. The embeddings derived from the graph attention network analysis were used for clustering to determine tumor cell villages. **B** UMAP of malignant cell groups based on the embeddings derived from graph attention networks in (**A**). **C** Village plots of different tumor cell villages. Dots represent malignant cells colored based on their cellular states. The size of each dot reflects the proportion of tumor cell states within a village. The thickness of the lines between dots indicates the strength of communication. Outer circles indicate the SDNs associated with each village, with different colored areas representing SDN proportions. **D** A

representative example (1CB) of tumor cell villages. Non-malignant cells were colored gray. **E, F, G, H** Top: Overall survival of liver cancer patients enriched in V1&2 related features (blue) and V3-8 related features (red) in a single-cell cohort (**E**, Ma et al., $n = 37$), as well as in the TCGA (**F**, TCGA-LIHC, $n = 365$), LCI (**G**, $n = 239$), and Mongolia (**H**, $n = 70$) cohorts. Kaplan–Meier plots were used to show the overall survival of each patient group. $p$-value was calculated using the log rank test. Bottom: Village feature score of the two patient groups in the top panels. Each box shows the median (center line), interquartile range (box), and data range (whiskers) of village scores. $p$-value was calculated using one-sided Student's $t$-test. ***, $p$-value $< 0.001$. Source data and detailed statistics are provided as a Source Data file.

determined by hierarchical clustering of tumors using surrogate village features. The observations align with previous studies showing both shared and distinct molecular features of HCC and iCCA[52]. Survival analysis further revealed that patients enriched for Villages 3–8 features had significantly poorer overall survival compared to those with features related to Villages 1 and 2 across multiple patient cohorts. These results suggest that tumor cell villages capture intrinsic tumor biology that is linked to patient prognosis in liver cancer. The defined tumor cell villages further allowed us to uncover unique molecular co-dependencies among cells within each village, which may play a critical role in orchestrating cell coordination and spatial organization. In silico perturbation of the identified gene pairs led to a significant decline in the accuracy of village identity prediction by the random forest model, highlighting their essential role in maintaining the integrity of tumor cell villages. Notably, *SPINK1*, derived from tumor cells, was spatially associated with genes expressed by mCAFs in Villages 3 and 6. SPINK1 is a serine protease inhibitor known for preventing the inappropriate activation of trypsin in the pancreas[53]. In liver cancer, *SPINK1* has been implicated in promoting tumor plasticity[54]. However, the interactions between *SPINK1*+ tumor cells and other cell types, as well as their roles in the spatial organization of tumors, remain poorly understood. Here, we demonstrate that *SPINK1*+ tumor cells may have unique molecular co-dependencies with mCAFs. Previous experimental studies have shown that SPINK1 can stimulate the proliferation of tumor cells and fibroblasts through EGFR signaling[46,47]. It is therefore likely that SPINK1 derived from tumor cells may also promote the proliferation of mCAFs via binding to EGFR. Perturbation of the molecular co-dependencies between *SPINK1*+ tumor cells and mCAFs leads to destabilization of the corresponding tumor cell villages. These findings suggest that the defined tumor cell villages reveal village-specific molecular co-dependencies that may contribute to shaping each tumor's unique spatial landscape, paving the way for developing strategies for therapeutic interventions.

This study has limitations. First, we performed single-cell spatial transcriptomic profiling and scRNA-seq on specimens from seven liver cancer patients (four HCC, three iCCA). While our analysis included over 2 million cells from 50 biospecimens sampled across multiple tumor regions, the findings may be constrained by the limited number of patients and the underlying molecular, histological, etiological, and demographic heterogeneity. Although we validated our observations using 10× Visium spatial transcriptome, single-cell transcriptome, and bulk transcriptome data, further validation in larger patient cohorts using tissue microarrays at single-cell spatial resolution would strengthen the tumor cell village concept. Second, molecular co-dependencies of individual tumor cell villages were determined based on correlations of gene pairs under spatial constraints. These gene pairs capture broader molecular relationships that are not limited to ligand-receptor interactions, and the paired genes may not directly interact with each other. While we used multiplex immunofluorescence and computational analyses to demonstrate the underlying mechanisms of *SPINK1* and its associated genes as a proof of concept, additional in vitro or in vivo studies are needed to fully

elucidate the cellular and molecular mechanisms of the identified gene pairs. Finally, this study focused only on liver cancer. Extending the tumor cell village concept to other cancer types may help further understand diverse tumor cell spatial organizations across malignancies.

## Methods
### Human sample collection
This study included seven liver cancer patients (four HCC, three iCCA; four males, three females) from the University Medical Center in Mainz and the NIH Clinical Center. Detailed information of this cohort can be found in Supplementary Data 1. The ages of the patients ranged from 63 to 77, with a median age of 72. A total of 50 samples were collected from these patients. We conducted single-cell spatial transcriptome profiling of 15 samples. Additionally, we generated single-cell transcriptome data for the remaining samples[20]. Sample collection was performed with informed consent from patients. This study was approved by the ethics committee of the University Medical Center in Mainz and the National Institutes of Health.

### CosMx SMI sample preparation
Formalin-fixed, paraffin-embedded (FFPE) tissue sections were prepared for CosMx SMI profiling following the procedures described in He et al.[19]. Briefly, five-micron tissue sections were cut and placed on Leica BOND PLUS slides. Slides were baked overnight at 60 °C to improve tissue adherence to the slide. Tissues were deparaffinized in Xylene twice for 5 min each. Then, tissues were successively dehydrated in 100% ethanol twice for 5 min each and subsequently dried at 60 °C for 5 more min. Tissues were then prepared for in-situ hybridization (ISH) by heat-induced epitope retrieval (HIER) at 100 °C for 15 min using 1× Target retrieval solution (NanoString CMx Slide Prep Kit, FFPE RNA 121500006) in a Bio SB cooker. After target retrieval, tissue sections were rinsed with diethyl pyrocarbonate (DEPC)-treated water (DEPC H₂O, ThermoFisher), washed in ethanol for 3 min and dried at room temperature for 30 min. Once the slides dried, an incubation frame was placed over the tissue (CosMx FFPE Slide Prep Kit RNA). Tissue was then digested with Proteinase K (provided by NanoString) 3 μg /ml at 40 °C for 30 min. Tissue sections were rinsed twice with DEPC H₂O, incubated in 1:400 diluted fiducials (Bangs Laboratories) in 2× saline sodium citrate and Tween (0.001% Tween 20, Teknova) for 5 min at room temperature, and washed with 1× PBS (ThermoFisher) for 5 min. After digestion and fiducial placement, tissue sections were fixed with 10% neutral buffered formalin (NBF) for 1 min at room temperature. Fixed samples were rinsed twice with Tris-glycine buffer (0.1 M glycine, 0.1 M Tris-base in DEPC H₂O) and once with 1× PBS for 5 min each before blocking with 100 mM *N*-succinimidyl (acetylthio) acetate (NHS-acetate, ThermoFisher) in NHS-acetate buffer (0.1 M NaP, 0.1% Tween PH 8 in DEPC H₂O) for 15 min at room temperature. The sections were then rinsed with 2× saline sodium citrate (SSC) for 5 min. NanoString ISH probes were prepared by incubation at 95 °C for 2 min and placed on ice, and the ISH probe mix (1 nM 980 plex ISH probe, 10 nM Attenuation probes, 1× Buffer R,

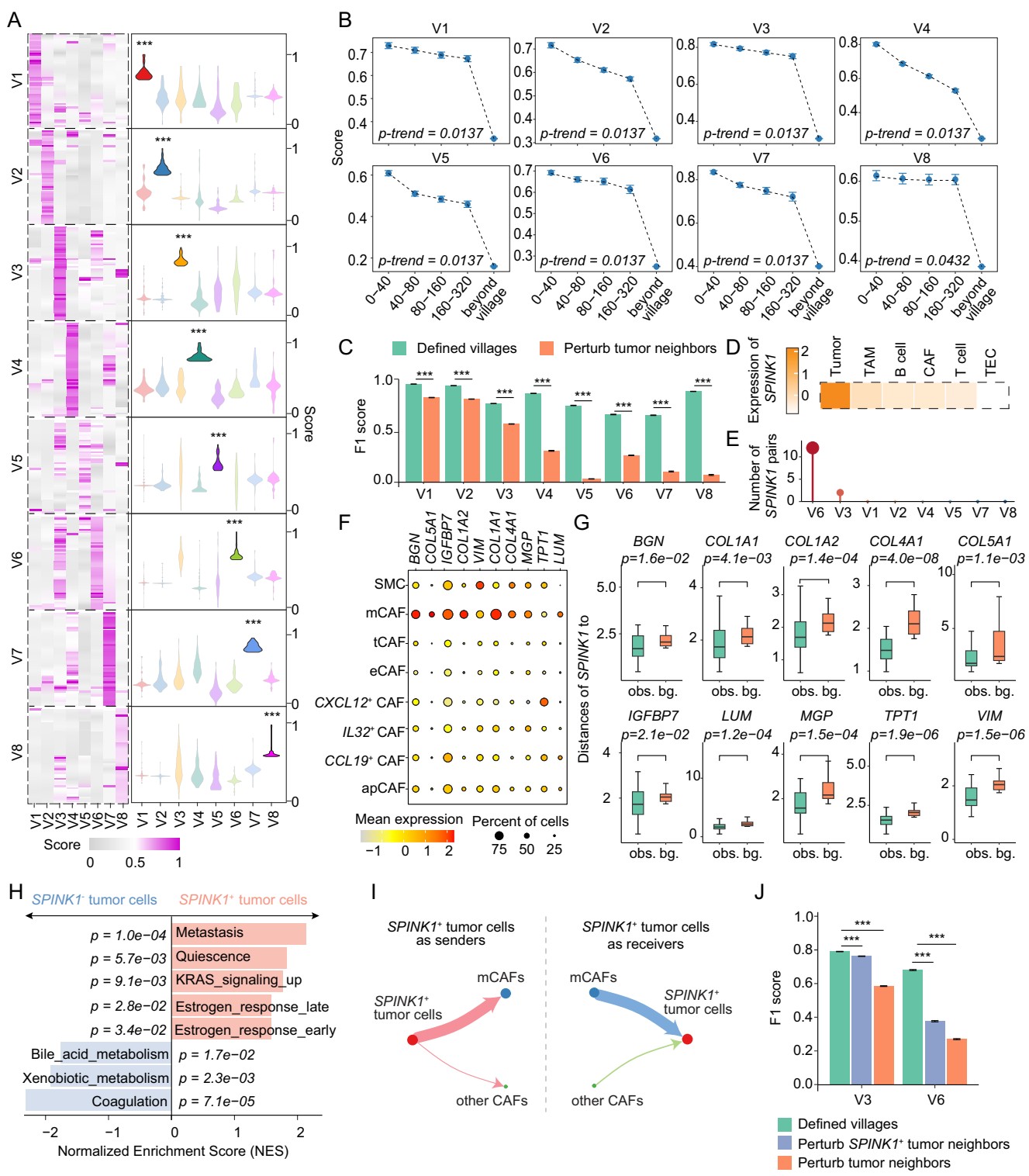

0.1 U/μL SUPERase•In™ [Thermofisher] in DEPC H$_2$O) was pipetted on the tissue slides. An adhesive SecureSeal Hybridization Chamber (Grace Bio-Labs) was placed over the tissue. The hybridization chamber was sealed to prevent evaporation, and hybridization was performed at 37 °C for 16–18 h. Tissue sections were rinsed of excess probes in 2× SCT for 1 min and washed twice in 50% formamide (VWR) in 2× SSC at 37 °C for 25 min, then twice with 2× SSC for 2 min at room temperature. Samples were then prepared for antibody cocktail incubation. First, slides were incubated with nuclear stain 1:40 (provided by NanoString) for 15 min at room temperature, protected from light.

After a short PBS wash, tissues were incubated with a 4-fluorophore-conjugated antibody cocktail against CD298/B2M (488 nm), PanCK (532 nm), CD45 (594 nm), and CD68 (647 nm) proteins and DAPI stain in the CosMx SMI instrument. Antibodies were also included in the CosMx slide preparation kit by NanoString. After 1 h incubation at room temperature protected from light, slides were washed three times with 1× PBS 5 min each. A custom-made flow cell was affixed to the slide in preparation for loading onto the CosMx SMI instrument. Flow cells were then stored in SSC while the instrument was being prepared for loading the samples.

**Fig. 6 | Spatial molecular co-dependencies in individual tumor cell villages.**
**A** Correlation scores of the top 50 gene pairs in each village. Violin plots (right) show the distribution of the scores of the gene pairs depicted in the heatmap (left) for each tumor cell village. Correlation scores were normalized to a 0–1 scale within each tumor cell village. The minimum, maximum, median, 25th percentile, and 75th percentile values for each violin plot are provided in the Source Data file. Statistical significance between each village's correlation scores and those of all other villages was assessed using one-sided Student's *t*-tests. Source data and detailed statistics are provided as a Source Data file. ***, *p*-value < 0.001. **B** Gradient changes in correlation scores between tumor cells and their neighbors across increasing spatial distances. Data are shown with mean ± standard error of the mean (SEM). Statistical significance of the trend was determined using the Mann-Kendall trend test. For each village, the top 50 gene pairs were included. Spatial bins represent increasing tumor cell-neighbor distances (0–40 μm, 40–80 μm, 80–160 μm, and 160–320 μm). Correlation scores were normalized to a 0–1 scale within each village. Source data and detailed statistics are provided as a Source Data file. **C** F1 score of the performance of the random forest model on the defined tumor cell villages and after perturbing the neighbors of tumor cells. Bar plots and error bars represent the mean F1 score + standard deviation (SD) (*n* = 100 replicates) for each tumor cell village. Each replicate represents an independent computational iteration of the random forest model. Statistical significance was assessed using one-sided Student's t-tests. No adjustment was made for multiple comparisons. ***, *p*-value < 0.001. Source data and detailed statistics are provided as a Source Data file. **D** Expression of *SPINK1* in different cell types. **E** Number of *SPINK1*-related gene

pairs in each tumor cell village. **F** Expression of genes across different fibroblast subtypes. Color represents average gene expression, while dot size indicates the fraction of cells expressing a certain gene. **G** Distances between *SPINK1*+ and specific marker positive (*BGN*, *COL1A1*, *COL1A2*, *COL4A1*, *COL5A1*, *IGFBP7*, *LUM*, *MGP*, *TPT1*, and *VIM*) spots shown as observations (obs.), compared to distances between *SPINK1*+ and randomly selected spots as background (bg.). Each box shows the median (center line), interquartile range (box), and data range (whiskers). Statistical significance was assessed using one-sided Wilcoxon rank-sum tests by comparing the observation and the background. Source data and the precise number of comparisons for each plot are provided as a Source Data file. **H** Gene set enrichment analysis of *SPINK1*+ tumor cells compared with *SPINK1*- tumor cells. *p*-values were calculated by permutation testing (*n* = 10,000 permutations). No adjustment was made for multiple comparisons. **I** Cell-cell interaction networks between *SPINK1*+ tumor cells (red) and mCAFs (blue) or other CAFs (green). Line width and dot size are scaled proportionally to the identified number of ligand-receptor pairs. **J** F1 score of the performance of the random forest model on the defined tumor cell villages, perturbing the neighbors of *SPINK1*+ tumor cells, and perturbing the neighbors of all tumor cells. Bar plots and error bars represent the mean F1 score + standard deviation (SD) (*n* = 100 replicates) for each tumor cell village. Each replicate represents an independent computational iteration of the random forest model. Statistical significance was assessed using one-sided Student's *t*-tests. No adjustment was made for multiple comparisons. ***, *p*-value < 0.001. Source data and detailed statistics are provided as a Source Data file.

## CosMx imaging
RNA target readout on the CosMx SMI instrument was performed according to the procedures described by He et al.[19] Briefly, the assembled flow cell was loaded onto the instrument, and Reporter Wash Buffer was flowed to remove air bubbles. A preview scan of the entire flow cell was taken, and fields of view (FOVs) were placed on the tissue to match regions of interest identified. In our case, we selected as many FOVs as needed to completely cover the whole tissue on the slide. RNA readout began by flowing 100 μl of Reporter Pool 1 into the flow cell and incubation for 15 min. Reporter Wash Buffer (1 mL) was flowed to wash unbound reporter probes, and Imaging Buffer was added to the flow cell for imaging. Eight Z-stack images (0.8 μm step size) for each FOV were acquired, and photocleavable linkers on the fluorophores of the reporter probes were released by UV illumination and washed with Strip Wash buffer. The fluidic and imaging procedure was repeated for the 16 reporter pools, and the 16 rounds of reporter hybridization-imaging were repeated multiple times to increase RNA detection sensitivity. Then, tissue and cell membrane morphology visualization was performed using oligonucleotide-conjugated antibodies hybridized to a specific pool of barcode reporters. After unbound antibodies and DAPI stain were washed with Reporter Wash Buffer, Imaging Buffer was added to the flow cell, and eight Z-stack images for the 5 channels (4 antibodies and DAPI) were captured.

## CosMx™ SMI data quality-control and preprocessing
With the coordinates and the raw transcript counts of individual cells, we filtered cells based on the following criteria: ≥20 counts detected and ≥10 genes detected; <10% negative probes and ≤5 negative probe counts. Cells with outlier cell area (*p*-value < 0.01) based on Grubb's test were also removed for downstream analysis. A total of 2,347,589 cells passed the quality control with a median of 93 features and 174 counts per cell. Data normalization was performed using the R package Seurat (v5.1.0).

## Major cell type annotation
Major cell type annotation (T cells, B cells, myeloid cells, endothelial cells, fibroblasts, and epithelial cells [including hepatocytes, cholangiocytes, and tumor cells]) of the single-cell spatial data was based on label transfer using the scRNA-seq dataset[20]. The R package Harmony (v1.2.0) was used to integrate the CosMx™ SMI data with the scRNA-seq data. The top 20 principal components (PCs) were used for

the integration, and the top 20 harmony embeddings were used to cluster the cells in the integrated dataset with default parameters. For each derived cluster, the percentage of different cell types from the scRNA-seq dataset was calculated. Clusters were annotated based on the highest proportion of cell types from scRNA-seq. A cluster was considered as unclassified if the highest proportion <50% or the ratio of the second-highest proportion against the highest proportion >50%. The analyses were performed for each sample separately. The expression score for each cell type was calculated as the mean values of the cell type-specific genes (Supplementary Data 2).

Malignant cells were determined based on their transcriptomes and geographic locations in a tumor. For epithelial cells from each individual patient, we identified different clusters of epithelial cells using Seurat. Specifically, after data normalization, most variable features were selected based on mean expressions and standard variances of genes. Top 20 PCA components were used for detecting neighbors, and Louvain clusters of epithelial cells were determined using default parameters. Each cluster was subsequently mapped to its spatial location. Clusters of epithelial cells located within tumor regions, based on the histological images, were classified as malignant cells, while those in non-tumor regions were annotated as non-tumor epithelial cells.

## Consistency of CosMx™ SMI and scRNA-seq data
To compare the gene expression profiles between CosMx™ SMI and scRNA-seq data generated from the same patients, we grouped the cells into six major cell types, including T cells, B cells, myeloid cells, endothelial cells, fibroblasts, and epithelial cells. We selected the genes that were detected in both datasets and calculated the average expression of each gene in individual cell types. We further calculated the Pearson correlation coefficient (R function "cor.test") between averaged gene expression profiles from CosMx™ SMI data and scRNA-seq data to measure the consistency of the two profiling approaches.

## Define tumor cell transcriptomic states based on gene modules
Due to the limited number of genes in the single-cell spatial data compared with scRNA-seq data, we determined gene modules in malignant cells based on scRNA-seq data and further applied those identified modules into the single-cell spatial data to define tumor cell transcriptomic states. Gene module identification was conducted based on a previous published non-negative matrix factorization

(NMF)-based method[5]. We included two liver cancer single-cell datasets to improve the robustness and sensitivity of gene module identification[15,20]. For each sample, the top 500 highly variable genes (HVGs) were used for data scaling and the NMF analysis. The number of cells was down-sampled to 1000 for the samples containing over 1000 tumor cells. Negative values were set to 0, and genes with expression values of 0 across all cells within each sample were discarded. We then performed NMF on the scaled gene expression matrix for each sample separately using the R package NMF (v0.27). The "nsNMF" method and "nndsvd" seeding method were applied. The rank was set to 10 if the number of malignant cells per sample was less than 400, and to 30 for samples containing 400 or more malignant cells. The contribution matrices of genes in different modules for each sample were obtained. A gene was assigned to a module if it showed the highest contribution to that module across all modules within a sample. Modules obtained from individual samples were further filtered by the Jaccard index. Gene modules with ≥5% overlap with at least two other modules were retained. For each paired gene, the number of individual tumor modules in which they co-existed (designated as the connection value) was calculated, and a gene–gene connection matrix was constructed for gene pairs. Genes with a connection value greater than three with at least four other genes were retained for downstream analysis. The graph was clustered using the infomap clustering method from the R package igraph (v2.0.3). The final modules with at least 5 genes were retained. Using this approach, we identified 11 modules across all samples. We annotated the gene modules based on gene ontology (GO) terms from MSigDB[55], tumor-derived signatures from Neftel et al.[56], Puram et al.[57], Ji et al.[58], and Barkley et al.[5], Hypergeometric distribution test was used to determine the enrichment of our identified consensus module genes in annotated gene sets with R function "phyper". FDR method was used to adjust the p-values.

Based on the gene modules identified from scRNA-seq data, module scores of individual epithelial cells from CosMx™ SMI dataset were calculated using the method from Barkley et al.[5], For each module, 10,000 randomized gene lists were generated, each containing the same number of genes as the module. The average gene expression for each random list and the module was then calculated. The score for the module was defined as:

$$score = -log10(\frac{n+1}{10000})$$

Where n represents the number of random gene sets with higher average expression than the module. The score was further linearly rescaled to between 0 and 1. We excluded "Metabolism" and "Apoptosis" modules from further analysis due to less than two genes detected in the CosMx™ SMI dataset. We removed cells with a score variance less than 0.15 and a mean score less than 0.2 across all modules to increase the confidence of module determination. The derived matrix was used to perform PCA and UMAP analyses. The epithelial cell clusters were determined by a graph-based method using the top 8 PCA components with default parameters in Seurat (v5.1.0). Average gene module scores were calculated for each cluster. Clusters with similar module scores were merged based on their hierarchical relationship, and a total of 32 epithelial clusters were obtained. Epithelial clusters were annotated to specific transcriptomic states based on the highest gene module scores.

## Cell subtypes in non-malignant cells

For each non-malignant cell type (myeloid cells, T cells, B cells, endothelial cells, and fibroblasts) in the single-cell spatial transcriptome data, Harmony (v1.2.0) was used to integrate cells from different patients. Top 20 PCA components were used to perform harmony analysis, and the top 20 harmony embeddings were used for UMAP and clustering. Clusters were annotated based on differentially

expressed genes with thresholds of adjusted p-value < 0.05 and log2 fold change ≥0.5. Marker genes from literature were used as reference in this analysis, including markers from Goveia et al.[27] for endothelial subtypes, from Ma et al.[29] for myeloid subtypes, from Cords et al.[59] and Zhang et al.[60] for fibroblast subtypes, from Morgan et al.[61], Yang et al.[62], and Zhang et al.[63] for B cell subtypes, and from Guo et al.[64], Zhang et al.[65], Zheng et al.[66], and Ma et al.[1] for T cell subtypes.

## Align cell subtypes between CosMx™ SMI data and scRNA-seq data

To compare cell subtypes identified from the CosMx™ SMI data and the scRNA-seq data, we performed correlation analysis of the clusters identified from the two datasets. Specifically, we conducted cluster analysis of each non-epithelial cell type, i.e., myeloid cells, T cells, B cells, endothelial cells, and fibroblasts, from the scRNA-seq data with top 10 PCA and default resolutions. We selected the common genes from the CosMx™ SMI and scRNA-seq data to calculate the Pearson correlation coefficients between each pair of clusters from the two datasets. A CosMx™ SMI cluster was considered present in the scRNA-seq data if its correlation coefficient with any scRNA-seq cluster exceeded 0.5.

## Characterization of the surroundings of individual malignant cells

To characterize the surroundings of individual malignant cells, we determined the composition of surrounding cells for each targeted malignant cell in the CosMx™ SMI data. The surrounding cells are defined as those located within a 40 μm radius of the targeted cell. We included three layers of information to characterize the surrounding cell compositions, including major cell types, cell states/subtypes, and cell clusters. This vector of cell compositions was used as features to describe the targeted cell. We further performed PCA analysis of the surrounding composition matrix for all malignant cells and used the top 10 PCA components to generate a UMAP representation with default parameters. Clusters of malignant cells were determined using the Louvain algorithm at a resolution of 0.6. Each cluster of malignant cells was annotated based on the composition of the surrounding cells.

To determine the distance for surrounding cell characterization, we tested the radii of 20, 40, 60, 80, and 100 μm. We performed pairwise comparison of the derived results using a normalized mutual information score. We also tested the significance of the comparisons based on a random shuffling strategy. We used 1000 times of randomly shuffled malignant cells as a background to calculate the p-value. Compared with 20 μm, malignant cell clusters are more stable with a larger radius. At a radius of 40 μm, the median number of surrounding cells for all malignant cells is 33. Considering that a larger radius will introduce a higher number of surrounding cells with a higher probability of noise to characterize a targeted cell, we selected a radius of 40 μm to determine the surroundings of individual malignant cells in our liver cancer single-cell spatial data.

## Enrichment of tumor cell states on SDNs

In the CosMx™ SMI data, the enrichment of tumor cell states on SDNs was calculated as the proportions of each tumor cell state with a specific SDN. We used a random shuffling approach (1000 times of randomly shuffling of tumor cell states) to calculate p-values and to determine the significance of the enrichment. To identify significantly enriched tumor cell states, p-values were calculated in a stringent way as p-value = n/1000, where n represents the number of cases with expected scores ≥ 0.8*observed score. A p-value less than 0.05 was considered significant.

We used 39 samples from three publicly available 10× genomics Visium datasets from, e.g., Liu et al.[38], Wu et al.[6], and Zhang et al.[39] to validate the enrichment of tumor cell states on SDNs. Among them, 24 samples from the tumor core were included in our analysis. To

determine tumor cell state for each spot, we generated reference pseudo-bulk gene signatures based on the CosMx™ SMI data. We averaged gene expression of the same tumor cell states to build an expression vector for each state. Genes with mean expression <0.1 or standard deviation <0.1 across all cell states were removed to retain the variable genes. We further selected the genes that were detected in the three datasets to ensure consistency. For each Visium dataset, Spearman correlations were then calculated between reference cell state signatures and the gene expression of each spot from the Visium data, resulting in a correlation matrix (Matrix I) of cell states and spots. High correlation values indicate a high probability of the spots containing tumor cells with the indicated cell states. To determine the SDNs for each spot in the 10× Visium datasets, we collected all cells related to each SDN in the CosMx data. The averaged gene expression related to each SDN was determined. Similar to the analysis of tumor cell states, we used the same criteria to filter genes and calculate correlations to determine the SDN of each spot. A correlation matrix (Matrix II) of SDNs and spots was generated. To infer the tumor cell state enrichment on SDNs, we calculated the Pearson correlation between tumor cell states (from Matrix I) and SDNs (from Matrix II) across all spots. The Pearson correlation coefficients were linearly rescaled to a range between 0 and 1.

## Physical distance between malignant cells to endothelial cells

For each malignant cell, the distance to the closest endothelial cell was calculated using the R function "kNN" from dbscan (version 1.1-12). R function "kdtree" was used to search the nearest cell.

## Identification of tumor cell villages

To identify tumor cell villages, we constructed an undirected graph for tumor cells from each sample, where nodes represent tumor cells and edges indicate connections of tumor cells within 40 μm distance. The cell state and the SDN were represented by one-hot encoding and combined as node features. Graph Attention Network (GAT) model implemented in PyTorch (v1.21.1) was applied to train the connected graphs for node clustering. In contrast to other methods, GAT utilizes attention mechanisms. During training, the model learns the importance of each neighboring node's features to the target node as attention coefficients for the target node. The coefficients are then normalized by the SoftMax function to sum to 1. We used this attention scoring as a representation of the interaction strength between nodes. These coefficients are used during feature aggregation in that the learned node feature is a weighted sum of neighbor information according to the attention coefficients. A 128-dimensional hidden representation was used for message passing between neighbors. Three convolutional layers was used in our analyses to aggregate neighborhood information of 3-hops. The Exponential Linear Unit (ELU) activation function was applied for each layer except for the last layer, followed by a 25% dropout rate for regularization. The model outputted a 15-dimension feature vector for each cell. To maximize the similarity of the embedded neighboring cells while minimizing that of distant cells, we used an evaluation method by Shiao et al.[67] Specifically, a minibatch of 10,000 target cells was randomly sampled from the input graph. For each target cell, we sampled two types of cells: one directly connected to the target cell, and another chosen from a random position in the graph, with the latter assumed to be spatially unrelated to the target cell. The evaluation function maximizes similarity between neighboring cells and minimizes similarity of unrelated cells. The Adam optimization algorithm was used during training with a learning rate of 1e-5 for 100 epochs. We performed dimensional reduction on the derived embeddings from GAT for all the tumor cells using PCA. The top 10 PCA components were used for Louvain clustering at a resolution of 0.5, resulting in 10 tumor clusters. Clusters 9 and 10 were discarded from the downstream analysis due to a small number of cells, accounting for less than 1% of total malignant cells.

## Village graph of malignant cells

Village graph represents the compositions of tumor cell states and SDNs associated with each village. The sizes of the dots indicate the proportion of tumor cell states in each village. Tumor cell states with a proportion of at least 5% were included in the graph. Interaction weights between tumor cells were obtained from the graph attention auto-encoder model described above, with higher values indicating stronger interactions. The interactions were visualized using the Fruchterman-Reingold layout. The outer ring stands for the SDNs, with different areas indicating the proportions of different SDNs related to each village.

## Define tumor cell villages using gene signatures

We determined gene signatures for each village based on differential gene expression analysis of the CosMx™ SMI data. Genes with fold changes ≥1 and p-values < 0.05 were selected as surrogate signatures for each village. To determine the specificity of the signatures to each village, we constructed a pseudo-bulk dataset based on the CosMx™ SMI data. For each tumor cell, surrounding cells within a 40 μm radius, measured by Euclidean distance, were collected. Average gene expression across cells was then calculated to generate pseudo-spot bulk gene expression. For each pseudo-spot, we determined the village scores as the mean expression of each village-related genes. Each pseudo-spot was assigned to the village with the highest village score. A confusion matrix was calculated between the predicted villages and ground truth from CosMx™ SMI data analysis. Accuracy, precision, recall, and F1 score were calculated to measure the performance of our village detection model. F1 score was calculated as follows:

$$F1 = 2 * \frac{precision * recall}{precision + recall}$$

For the 10× Visium dataset, we used the same method to determine tumor cell villages for the spots. For each dataset, village scores were then linearly rescaled to a range between 0 and 1 across all spots. The confidence of village assignment for each spot was calculated as the difference between the highest village score and the second-highest village score. For the scRNA-seq data, pseudo-bulk gene expression for each sample was determined, followed by tumor cell village score calculation.

## Overall survival

For each cohort, tumor samples were clustered based on the village scores. Hierarchical relationship of the samples was constructed using Pearson correlation as distance and "complete" agglomeration method. Samples in each cohort were grouped into two clusters based on hierarchical relationships. Kaplan-Meier plots were generated and visualized using "Surv" function from R package Survival (v3.7-0).

## Cell-cell interactions based on single-cell spatial transcriptome data

To infer cell-cell communications from the CosMx™ SMI data, we used the COMMOT method[68], with CellChatDB as the ligand-receptor (LR) database. For each sample, we included only LR pairs identified in ≥100 cells or in ≥5% of all tested cells. Cells within a 50 μm distance were considered for interaction calculations. To identify LR interactions associated with tumor cell states, we used a multi-step approach. First, LR scores were calculated for each tumor cell state using tl.cluster_communication function in COMMOT, with cells categorized into B cells, T cells, endothelial cells, epithelial cells, fibroblasts, myeloid cells, and 12 distinct tumor cell states. LR scores were considered significant if p-value < 0.05, determined by 1000 cell label shuffles. LR scores were set to 0 for non-significant interactions and further averaged across samples to minimize sample bias. Next, LR scores between different tumor cell states were compared; pairs with a log2-fold

change ≥0.5 and a signal in ≥10% of tumor cell states were selected. Finally, for each tumor cell state, a ranking score (score = P × FC) was calculated for each LR interaction based on the proportion of cells with the LR interaction (P) and the log2-fold change compared with other tumor cell states (FC), both rescaled between 0 and 1. The top three LR interactions for each tumor cell state were selected based on these ranking scores.

## Molecular co-dependencies in tumor cell villages

We developed a bioinformatic method to determine the molecular co-dependencies between tumor cells and their neighboring non-tumor cells (including CAFs, TAMs, TECs, T cells, and B cells) within each tumor cell village. Specifically, for a given tumor-neighboring cell type pair in a tumor cell village, we identified the top 50 highly variable genes (HVGs) for each cell type. For each tumor cell-neighboring cell type pair, we randomly selected 1000 tumor cells, 1000 paired neighboring cells of a given cell type located based on a 40 μm distance. Pearson correlation coefficients were calculated between each pair of tumor-HVGs and non-tumor cell HVGs (each cell type separately). We repeated the random selections for 1000 times. The top 50 gene pairs with the highest correlations were selected for each tumor cell village. We further determined the correlations of these gene pairs within each tumor cell villages based on 40–80, 80–160, and 160–320 μm distance, and beyond each tumor cell village. The correlation scores were further linearly rescaled to a range between 0 and 1 within each village.

## Random forest model to assess molecular co-dependency

We used a random forest model to evaluate the importance of the identified gene pairs in each tumor cell village. A random forest model was built using expression profiles of the identified gene pairs. For each tumor cell, the expression of the paired genes in a corresponding neighboring cell type was averaged based on a 40 μm distance. We used two-thirds of the tumor cells in each village as the training set, with default parameters from the R package random-Forest (v4.7-1.2). Tumor cell villages were used as classifier labels. The remaining one-third of the tumor cells served as the test set to evaluate model performance. To examine the importance of the gene pairs, we spatially shuffled the neighbors of tumor cells to construct perturbed neighboring expression profiles and assess the model's performance on these perturbed profiles. Additionally, we spatially shuffled the neighbors of *SPINK1*+ tumor cells to specifically evaluate the impact of the *SPINK1*-related gene pairs in Villages 3 and 6. The model-building and neighbor perturbation steps were repeated 100 times to mitigate potential random effects and ensure robustness of the results.

## Spatial co-localization of *SPINK1*+ spots and spots positive for given genes

For each sample in the 10X Visium dataset, we identified *SPINK1*+ spots as those with the top 25% *SPINK1* expression across all spots. Genes of interest (GOI, e.g., *BGN*, *COL1A1*, *COL1A2*, *COL4A1*, *COL5A1*, *IGFBP7*, *LUM*, *MGP*, *TPT1*, and *VIM*)-positive spots were similarly defined using the top 25% expression threshold. For each *SPINK1*+ spot, we used the "kNN" function from the R package dbscan (v1.2-0) to identify the nearest GOI-positive spots and calculate their distances. The distances for all *SPINK1*+ spots were averaged as the observed distance between *SPINK1*+ spots and GOI-positive spots for each sample. To assess significance, we generated 1000 random spot lists, each containing the same number of spots as GOI-positive spots. For each random list, "kNN" was used to calculate the average distance of *SPINK1*+ spots and the random spots (random distance). *P*-values were calculated as $p = n/1000$, where $n$ is the number of random distances equal to or less than the observed distance for the given gene pair in a sample.

## Gene set enrichment analysis (GSEA)

R package fgsea (v1.32.0) was used to perform GSEA analysis. Pathways with *p*-values < 0.05 were considered as significantly enriched pathways. For single sample GSEA (ssGSEA) analysis, R package GSVA (v2.3.1) was used. Pathways were obtained from the hallmark gene sets of GSEA database. Pathways with enrichment scores ≥ mean + standard deviation of all derived scores were retained.

## Cell–cell communications based on single-cell transcriptome data

The function "nichenet_seuratobj_cluster_de" from the R package NicheNetR (v2.1.0) was used with default parameters to infer cell-cell communication in single-cell transcriptomics. For each of the identified ligands, the top-weighted receptor was selected from the Niche-Net database. Ligand-receptor pairs were retained if the receptors (tumor cells as senders) or the ligands (CAFs as senders) were significantly differentially expressed between mCAF and other CAFs (adjusted *p*-value < 0.05). *p*-values for ligands were calculated using two-sided Wilcoxon test and corrected for multiple testing using the Benjamini–Hochberg false discovery rate.

## Multiplex immunofluorescence

Snap-frozen CCA tissue was embedded in a tissue freezing medium (Leica) and cut into 5 μm sections. The sections were then stained using a standard immunofluorescence protocol. Primary antibodies against SPINK1 (1:50, Invitrogen, Cat # PA5-102353) and COL1A1 (1:80, Cell Signaling, Cat # 66948S) were used. The corresponding secondary antibodies conjugated with fluorophores (anti-rabbit Alexa 488, 1:500, Cat # 4412S; anti-mouse Alexa 555, Cell Signaling, Cat # 4409S) were then added. Counterstaining was performed with DAPI (1:1000, Carl Roth). The fluorescently stained tissues were viewed with a Leica DMI8 confocal microscope using a 63x magnification objective with the Tile Scan function, and the images were processed using LAS X Office software.

## Statistics and reproducibility

No statistical method was used to predetermine sample size. No sample in this study was excluded from the analysis. The experiments were not randomized. The investigators were not blinded to allocation during experiments and outcome assessment.

## Reporting summary

Further information on research design is available in the Nature Portfolio Reporting Summary linked to this article.

# Data availability

The single-cell spatial transcriptome data generated from this study have been deposited in Zenodo (https://zenodo.org/doi/10.5281/zenodo.13773977). The processed scRNA-seq data of the patients in this study are available through the Gene Expression Omnibus (accession number GSE189903). Raw sequencing data are considered protected information and are therefore available under restricted access through dbGaP under accession number phs003117.v1.p1. Access via the NCI's dbGaP can be requested by qualified senior or principal investigators overseeing the research. The NCI's Data Access Committee reviews such requests within 3 months and will make data available for up to 12 months. The publicly available 10X genomics visium datasets used in this study include samples from Liu et al.[38] (Mendeley Data: skrx2fz79n, https://data.mendeley.com/datasets/skrx2fz79n), Wu et al.[6] (http://lifeome.net/supp/livercancer-st/data.htm), Zhang et al.[39] (GEO accession: GSE238264, https://www.ncbi.nlm.nih.gov/geo/query/acc.cgi?acc=GSE238264), and Mo et al.[42] (HTAN DCC Portal under the HTAN WUSTL Atlas, https://data.humantumoratlas.org/). Other publicly available data include bulk transcriptomic data of GSE14520 (LCI)[43], GSE144269 (Mongolia,

https://www.ncbi.nlm.nih.gov/geo/query/acc.cgi?acc=GSE144269)[44], and the TCGA database (TCGA-LIHC, https://portal.gdc.cancer.gov) and scRNA-seq data of GSE151530, https://www.ncbi.nlm.nih.gov/geo/query/acc.cgi?acc=GSE151530[1].

## Code availability

Code used in this project have been deposited in github (https://github.com/MengLiu1/Tumor-cell-villages).

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

## Acknowledgements

We thank Drs. Xin Wei Wang, Eytan Ruppin, and Tom Misteli for helpful comments on the manuscript; Dr. Yuuki Ohara for assistance in interpreting the histology images; the patients, families, and nurses for contribution to this study. This work was supported by grants (ZIA BC 012079 [L.M.] and ZIA BC 012083 [L.M.]) from the intramural research program of the Center for Cancer Research, National Cancer Institute of the United States. J.U.M. is supported by grants from the Wilhelm Sander Foundation (2021.089.1). The contributions of the NIH authors were made as part of their official duties as NIH federal employees, are in compliance with agency policy requirements, and are considered Works of the United States Government. However, the findings and conclusions presented in this paper are those of the authors and do not necessarily reflect the views of the NIH or the U.S. Department of Health and Human Services.

## Author contributions
L.M. developed study concept; J.U.M. directed clinical study; M.L. performed computational analysis; M.O.H. conducted experiments; D.C., H.P.L., W.W., L.W., M.F., and J.M.H. conducted additional experiments and data analysis; M.L. and L.M. interpreted data; L.M. and M.L. wrote the manuscript with help from M.O.H. and D.C. All authors read, edited, and approved the manuscript.

## Funding

## Competing interests
The authors declare no competing interests.
