## [Transparent Peer Review file · Nature Communications]

Tumor cell villages define the co-dependency of tumor and microenvironment in liver cancer

Corresponding Author: Dr Lichun Ma

Version 0:

Reviewer comments:

Reviewer #1

(Remarks to the Author)

Liu et al. investigated the spatial organization of tumor cells and their microenvironment in primary liver cancer by integrating spatial single-cell imaging and single-cell RNA sequencing. They developed a deep learning-based strategy termed Spatial Dynamics Network to spatially map tumor cell states and their surrounding architectures. With these new results, they introduced the new concept of "tumor cell villages", which exhibited village-specific molecular co-dependencies between tumor cells and their microenvironment and were associated with patient outcomes. This concept highlights a centric role of tumor in tumorigenesis but also considers the critical contribution and diverse phenotype of immune and stromal cells. Overall, this paper is technically sound and potentially a great resource for liver cancer. The new tumor village concept is interesting and offers a new angle to understand tumor heterogeneity. There are several concerns needed to be addressed before publication.

Major:

1. The study includes both HCC and iCCA patients and notes specific enrichment patterns for tumor cell states and villages between these subtypes. Given the distinct biology and clinical management of HCC and iCCA, it would be beneficial to more explicitly discuss whether certain villages or co-dependencies are predominantly found in one subtype versus the other and the potential clinical implications.

2. The features of Villages 1 and 2, specifically a vascularized environment, the EMT state, and enrichment of iCAF, are well-established indicators of poor tumor prognosis. Why do patients enriched for V1 and V2 show a better prognosis compared to other patients (Figure 5E)?

3. Figure S7B, derived from the primary CosMx SMI dataset, suggests that many of the identified tumor cell villages exhibit a largely patient-specific distribution, often being prominent in only a single patient or a small subset of patients within the initial cohort. However, when the derived village gene signatures are applied to validation cohorts (as indicated in analyses of 10X Visium data, Figure S7G/H), a different pattern seems to emerge where a majority of patients/samples appear to be predominantly composed of signatures related to Village 1, Village 2, and Village 8.

These two sets of observations do not seem to fully align. Could the authors please clarify this apparent inconsistency? Is this discrepancy primarily attributable to the inherent differences and resolutions between the various technology platforms? Alternatively, could it suggest potential limitations in the stability, robustness, or generalizability of the identified village gene signatures when extrapolated to more diverse patient datasets?

4. A mIF staining of SPINK1+ tumor cells is needed to validate the existence and the surrounding TME architecture of these malignant cells.

5. The findings, like spatial preference of tumor cell state, were validated in published 10X Visium spatial transcriptome data from 39 HCC patients, which is comparable to the discovery cohort. However, the bulk RNAseq data used for validation lacks single cell resolution. A much larger single cell RNA-seq data may be more appropriate to validate the findings.

6. How to explain the clinical relevance of the identified tumor villages since multiple villages are observed in one patient?

7. The tumor niche or microenvironment is a well-established concept to describe the tumor cell and surrounding immune cells, etc. A special paragraph is needed to discuss the similarities and differences between the tumor niche and tumor village concepts to help the readers understand this proposed new concept.

Minor:

1. For the reader's convenience, within the same figure (for example, Figure 2C), the same cell type should be marked with the same color. This will make it easier for the reader to match the cell's name with its position in the UMAP plot.
2. Clustering of both tumor cells and non-tumor cells are not very clear to separate distinct cell types. There's a concern that batch effects in the spatial single-cell data may have been over-corrected during processing, potentially affecting the clustering result.
3. In the abstract section, it is the tumor village, not the Spatial Dynamics Network, which is obtained through the deep learning-based strategy, as referred to the Methods sections.
4. Typo: page 29, "recal" should be "recall".

Reviewer #2

(Remarks to the Author)

In this manuscript, Liu et al have performed spatial transcriptomics and single cell RNA sequencing to interrogate the tumour heterogeneity, focussing on primary liver tumours. They documented that tumours are composed by distinct units constituted by different cell subtypes. In particular, different tumour cell state are preferentially associated with specific spatial dynamics networks. Collectively, results indicate a close relationship between tumour cell states and the cell subtypes composing their microenvironment. Outcomes propose a functional spatial context in which cell types composing each unit may mutually influence their support and behaviour. The authors introduced the concept of tumour cell "villages", as distinct cell subtypes may be linked to a reciprocal coexistence to support their entity, fitness, identify, and dependency. The functional relevance of these "villages" is exemplified by showing correlations between the presence of some villages and patient outcomes. Moreover, the authors exemplified a putative functional relationship by focussing on a gene, SPINK1, which expression in cancer cells of a given village appears to confer co-dependency with mCAFs.

The manuscript is very well structured, and the idea of tumour cell "villages" is very provocative and fascinating. Findings contribute to functionally understand and interpret inter- and intra- tumour heterogeneity, its relevance in supporting distinct cancer cell subtypes, and a putative vulnerability in tumour heterogeneity, which could be exploited through strategies designed to perturb the strength of distinct "villages".

In my opinion there is one major improvement needed to ensure clarity and avoid misinterpretations:

Since the beginning, and already in the abstract, the authors speak about 50 samples. It is then going through the text, checking details in supplementary information, that readers then realize that in fact these 50 samples correspond to 7 patients. In the present version of the manuscript, this information is not evidently reported, as it appears only in Table S1. The number of analysed patients must be clearly stated since the beginning. Moreover, the authors stated that they performed studies on primary liver cancers. In fact, they used 3 iCCA and 4 HCC. iCCA and HCC have distinct molecular, histological, vulnerable features. Moreover, male and female HCC have often very distinct characteristics. Consequently, the study is based on a small fraction of patients. Considering the complexity of the study and the deep analyses performed, I do not find that 7 patients are not sufficient. The authors should clearly state these issues and clarify the samples on which the study has been performed, to avoid misunderstanding and misinterpretations from readers. I recommend as well to discuss these limitations in the discussion sections, in addition to a pertinent section in which the authors have already discussed some limitations of their study. I also want to stress that, despite the number and diversity of samples analysed could be a limitation of the study, the main findings and concept of 'villages' are supported by their existence across different primary liver cancer, and notably in relation the complexity and heterogeneity of liver cancer. Moreover, the authors wrote in the result section: "We noticed that tumor cell village compositions were much more similar across different sampling regions from the same patient compared to those from different patients, suggesting relatively stable tumor cell populations within individual patients". As I mentioned above, this may be again related to the comparison of patients with distinct primary liver cancer types (HCC and iCCA) and from male versus female patients. The authors should widen their interpretation, as the one currently proposed would be only supported by analysing several patients belonging to the same primary liver cancer type and sex of patients. This additional aspect could be as well elaborated in the discussion section. To help readers interpreting differences identified in studies, I recommend to reports in all panels in figures and supplementary figures, which samples belong to HCC and to iCCA patients.

Minor comments:

- 1) Genes should be in *Italic*.
- 2) in S1, "B" should be positioned higher to make evident for readers that "fraction of cells" and "average expression" scales are referred to graphs below.

Reviewer #3

(Remarks to the Author)

In this manuscript, the authors collected a total of 50 samples containing tumor regions, boundary regions, and normal regions from tumor tissues of 3 iCCA patients and 4 HCC patients and studied intratumoral heterogeneity of primary liver

cancer using CosMx and scRNA-seq technologies. First, the authors classified and annotated cancer cells and non-cancer cells. Second, the authors calculated the number of non-tumor cells within a 40 μm range around each tumor cell, used this information for cancer cell clustering, and defined the results as Spatial Dynamic Networks (SDNs). Combining cancer cell states and SDNs, the authors constructed tumor villages using deep learning neural networks and found that Village 1 and Village 2 contained mixed tumor cell states and vascularized microenvironments, which were associated with good prognosis. Finally, the authors explored the molecular co-dependencies maintaining tumor village stability. From my perspective, the intention of the study is meaningful, but some conclusions in the article are not solid, with specific issues as below.

Major concerns:

- 1) In Figure 1A, the authors described multi-point tumor tissue sampling from the cancer region, boundary zone, and adjacent normal area for each patient. However, the paper provides minimal discussion on the differences among these three regions and the spatial variations within the cancer region itself. An in-depth analysis of these differences might better reveal intratumoral heterogeneity and address actual biological and clinical problems. This would be more meaningful than merely optimizing existing computational methods or showcasing fancier computational approaches.
- 2) The tumor cell states obtained through NMF in Figure 2A contain multiple stress states. The first question is, given that Stress/Metal and Stress are adjacent, why separate them instead of combining them into a single Stress state? The second question is, the stress comprised of four clusters shows results in Figure S2E that are mainly concentrated in the 2CB samples. Is this a sample-specific phenomenon? Including such information may not be conducive to explaining universal phenomena. The authors should filter out patient- or sample-specific states before proceeding with subsequent analysis.
- 3) Can the numerous cell subtypes identified in Figure 2C be mapped to the cell subtypes currently recognized in the pan-cancer cell atlas?
- 4) As shown in Figure 5C, the authors ultimately identified 8 tumor villages. In Figure S7B, the authors demonstrated the tumor village composition of each sample, showing that V1 is a village specific to patient 2H, while V2 is specific to patient 3H. Looking at the SDNs in Figure 5C, T3, 4, and 5 are specific to V1, while T1 and 2 are specific to V2. Since I couldn't find the composition of T across samples, I can only speculate whether V1 (T3, 4, 5) and V2 (T1, 2) respectively represent spatial patterns specific to patients 2H and 3H? If this is indeed the case, it seems unreasonable to combine these two patient-specific tumor villages into one group for prognostic analysis.
- 5) Furthermore, according to the description, both V1 and V2 belong to a vascularized environment. Theoretically, an environment associated with vascularization should be related to tumor metastasis and consequently linked to poor prognosis. However, the authors' research results show that this is associated with good prognosis - how can this phenomenon be explained?
- 6) The author introduces the concept of molecular co-dependencies at the end of the article, identifying multiple gene pairs through correlation analysis and suggesting these gene pairs are factors that maintain village stability. However, I notice that these gene pairs are not ligand-receptor pairs. If the genes within these pairs cannot directly interact with each other, then the claim that such pairing can stabilize the village seems rather far-fetched. This correlation is more likely due to spatial proximity, leading to related phenotypes in their respective cell types. Therefore, rather than attributing it to the action of gene pairs, it would be more accurate to say that the coexistence of two cell types facilitates the formation of villages. I believe that according to the concept proposed by the author, the gene pairs identified through analysis need to be explained through molecular and cellular mechanisms, which currently seem weak and require more direct evidence.

Minor concerns:

- 1) Please label the sample size information from Table S1 in Figure 1A. While the article studies intratumor heterogeneity, the schematic diagram creates confusion by showing only one sample each from tumor, boundary, and normal tissue.
- 2) While I understand the author's intention to make the visualization more prominent, the scores in the six graphs in Figure 1D have been adjusted too artificially and no longer appear to be continuous values.
- 3) The authors present multiple clustering results in the paper, all simply named c1, c2, etc. Although Figure S4 indicates the subtype classification of each cell type, Figure 2A does not clearly specify that these are tumor subtypes. It is suggested to add abbreviations of the major categories before c to avoid confusion between different clustering results.

Version 1:

Reviewer comments:

Reviewer #1

(Remarks to the Author)

All my concerns have been addressed. I have no further comments.

Reviewer #2

(Remarks to the Author)

The authors have adequately addressed the points raised. Congratulation for this inspiring work.

Reviewer #3

(Remarks to the Author)

The authors have basically answered all the questions. However, I still have a minor question about the response to the first comment. Regarding the epithelial cells in the Supplementary Figure 6A, according to the description in line 579 of the

manuscript, these cells include hepatocytes, cholangiocytes, and tumor cells. Among them, normal epithelial cells should be enriched in the adjacent non-tumor area, while tumor cells should be enriched in the tumor area. Therefore, observing the regional enrichment pattern of total epithelial cells, which include both normal and cancerous epithelial cells, seems unreasonable.

Version 2:

Reviewer comments:

Reviewer #3

(Remarks to the Author)

I have no further comments.

Responses to Reviewers' Comments

We would like to express our sincere gratitude to the reviewers for their constructive and helpful comments to improve our paper. Below are the point-by-point responses to the reviewers.

Reviewer #1:

Major comments:

1. The study includes both HCC and iCCA patients and notes specific enrichment patterns for tumor cell states and villages between these subtypes. Given the distinct biology and clinical management of HCC and iCCA, it would be beneficial to more explicitly discuss whether certain villages or co-dependencies are predominantly found in one subtype versus the other and the potential clinical implications.

Response: Thank you for this helpful suggestion. We fully agree with the reviewer that HCC and iCCA represent two clinically distinct subtypes of liver cancer, each with characteristic histological features and clinical management strategies. At the same time, prior studies have demonstrated that these subtypes may share certain molecular features (Chaisaingmongkol et al. *Cancer Cell* 2017, PMID: 28648284). In line with the reviewer's suggestion, we have expanded the Discussion to more explicitly elaborate on tumor cell villages in HCC and iCCA, the underlying molecular co-dependencies, and their potential clinical implications. In our analysis of single-cell spatial transcriptomic data from the discovery cohort, we found that while most tumor cell villages were detectable in both HCC and iCCA, certain villages may be preferentially enriched. Given the limited number of patients in this cohort, we further examined village enrichment using a larger single-cell transcriptome dataset that included both HCC and iCCA patients. This analysis showed that Villages 1 and 2 were preferentially enriched in HCC, whereas Villages 3–8 were observed in both HCC and iCCA, as determined by hierarchical clustering of tumors using surrogate village features. These results are consistent with prior evidence that HCC and iCCA have both shared and distinct features (Chaisaingmongkol et al. *Cancer Cell* 2017, PMID: 28648284; Ma et al. *Nat Commun*, 2022, PMID: 36476645). Survival analysis further revealed that patients with Villages 1 and 2-enriched tumors had better overall survival outcomes than those enriched for Villages 3–8. Consistent results were observed when the analysis was restricted to HCC patients alone and further validated in three patient cohorts with bulk transcriptome data. We didn't perform survival analysis on iCCA alone in the single-cell cohort, as all iCCA samples in this dataset were enriched for Villages 3–8. We also determined the molecular wiring underlying individual tumor cell villages by identifying correlated gene pairs. Among them, critical correlations between *SPINK1* from tumor cells and genes from mCAFs were uncovered in Villages 3 and 6, which were linked to poor patient outcomes. This observation was further supported by cell–cell interaction analysis and validated using multiplex immunofluorescence staining. Moreover, in silico perturbation of *SPINK1* and its paired genes resulted in significant decline of village identity prediction accuracy compared with the original village configurations. These results highlight the important role of

these gene pairs in maintaining the integrity of their corresponding tumor cell villages and pave the way for developing new treatment strategies aiming at disrupting these co-dependencies to destabilize the tumor. We have revised the manuscript accordingly. Please refer to Figures 5E-5H, 6I, 6J, S11C-S11E, S12B, and the Discussion section.

2. The features of Villages 1 and 2, specifically a vascularized environment, the EMT state, and enrichment of iCAF, are well-established indicators of poor tumor prognosis. Why do patients enriched for V1 and V2 show a better prognosis compared to other patients (Figure 5E)?

Response: We appreciate the reviewer for this insightful comment. To investigate this question, we performed gene set enrichment analysis and further examined specific tumor and microenvironmental features linked to the identified tumor cell villages. We found that Villages 1 and 2 were enriched in metabolic-related pathways, such as bile acid metabolism, adipogenesis, and fatty acid metabolism. In contrast, pathways such as MYC targets, E2F targets, G2/M checkpoint, mitotic spindle, reactive oxygen species pathway, PI3K/AKT/mTOR signaling, and angiogenesis were observed across multiple Villages within 3–8, corresponding to programs that promote cell growth, survival, resistance to cell death, and neovascularization. Notably, while Villages 1 and 2 contained more endothelial cells, the endothelial cells in Villages 3–8 exhibited significantly higher expression of tumor-specific endothelial signatures (derived from Croix et al. Science 2000, PMID: 10947988), consistent with the angiogenesis pathway observed in several of these villages. Previous single-cell studies have demonstrated that endothelial cells in cancer are heterogeneous, with some subtypes resembling normal vasculature and others displaying tumor-promoting properties (Li et al. Natl Sci Rev 2024, PMID: 39345334). Our results suggest that the endothelial cells in Villages 3–8 acquire more tumor-associated features compared with those in Villages 1 and 2, which may underlie the poorer prognosis observed in Villages 3–8. In our initial classification of CAFs, we defined subtypes based on clustering analysis and the differentially expressed genes in each cluster, following previously reported classifications of fibroblasts in the literature (Cords et al. Nat Commun 2023, PMID: 37463917). Specifically, clusters enriched for inflammatory-related genes were annotated as iCAFs (*CXCL12*), rCAFs (*CCL19*, *CCL21*, *CCL2*), and ifnCAFs (*IL32*, *CXCL9*, *CXCL10*). After considering the reviewer's comment, we revisited the literature and found that the defined rCAFs and ifnCAFs in our study are generally considered as subpopulations within the broader iCAF category, which we had initially overlooked (Lavie et al. Nat Cancer 2022, PMID: 35883004). To improve clarity and avoid confusion, we have therefore renamed these fibroblast subtypes according to their dominant marker genes: iCAFs as *CXCL12*+ CAFs, ifnCAFs as *IL32*+ CAFs, and rCAFs as *CCL19*+ CAFs. The total abundance of the three CAF subtypes did not differ significantly between Villages 1–2 and Villages 3–8. With respect to EMT, although a small proportion of EMT-like cells were detected in Village 2, EMT pathways were not significantly enriched, suggesting that these cells constitute only a minor fraction of the tumor and may not represent the dominant ecosystem state. These findings further underscore the complex interplay within the tumor ecosystem and highlight the value of the tumor cell village

concept for understanding how coordinated tumor cell states and their surrounding environments collectively shape overall tumor fitness and clinical outcomes. We thank the reviewer again for raising this question and we have revised the manuscript to include new analysis results. Please refer to Figures 2C, S3, S11F-11H, as well as the corresponding revised text in the main manuscript.

3. Figure S7B, derived from the primary CosMx SMI dataset, suggests that many of the identified tumor cell villages exhibit a largely patient-specific distribution, often being prominent in only a single patient or a small subset of patients within the initial cohort. However, when the derived village gene signatures are applied to validation cohorts (as indicated in analyses of 10X Visium data, Figure S7G/H), a different pattern seems to emerge where a majority of patients/samples appear to be predominantly composed of signatures related to Village 1, Village 2, and Village 8. These two sets of observations do not seem to fully align. Could the authors please clarify this apparent inconsistency? Is this discrepancy primarily attributable to the inherent differences and resolutions between the various technology platforms? Alternatively, could it suggest potential limitations in the stability, robustness, or generalizability of the identified village gene signatures when extrapolated to more diverse patient datasets?

Response: We appreciate the reviewer for the comment. As noted by the reviewer, one contributing factor is the resolution limitation of the 10x Visium platform, in which each spot measures 55 μm in diameter and contains a mixture of cell types. This inherent averaging effect reduces the ability to resolve fine-grained cell–cell relationships and accurately map tumor cell villages at true single-cell resolution, as achieved with CosMx SMI. Additionally, our initial 10x Visium analysis included only three HCC datasets available at the time, in which we observed enrichment of Villages 1, 2, and 7 (likely perceived by the reviewer as Village 8 due to color coding; we have modified the colors to improve readability), consistent with our CosMx-based findings. To expand validation, we have now incorporated 10x Visium data from an iCCA cohort reported in a recent publication (Mo et al. Nature 2024, PMID: 39478210). Consistent with the CosMx results, we observed that certain tumor cell villages, including Villages 3, 5, and 6, were enriched. These results support the robustness of the identified tumor cell villages in our study. The new data are presented in Figure S11B.

4. A mIF staining of SPINK1+ tumor cells is needed to validate the existence and the surrounding TME architecture of these malignant cells.

Response: Thank you for this helpful suggestion. Accordingly, we validated the colocalization of *SPINK1*+ tumor cells and mCAFs by performing co-staining of SPINK1 and COL1A1 (the most highly expressed mCAF marker). Consistent with our spatial analysis, we observed strong colocalization of SPINK1 and COL1A1 in tumors enriched for Village 6, where SPINK1+ tumor cells and mCAFs are closely associated. The data are presented in Figure S12B.

5. The findings, like spatial preference of tumor cell state, were validated in published 10X Visium spatial transcriptome data from 39 HCC patients, which is comparable to the discovery cohort. However, the bulk RNAseq data used for validation lacks single cell resolution. A much larger single cell RNA-seq data may be more appropriate to validate the findings.

Response: We appreciate the reviewer's helpful suggestion and agree that bulk transcriptome data offers limited resolution. Accordingly, we performed additional validation of our findings using a single-cell RNA-seq dataset from 37 HCC or iCCA patients (Ma et al. J Hepatol 2021, PMID: 34216724). Consistent with our initial results, patients enriched for Villages 3–8 had significantly poorer outcomes compared with those enriched for Villages 1 and 2. We further repeated the analysis in HCC patients alone and observed a similar trend. We didn't analyze iCCA separately because all iCCA tumors in this single-cell cohort were enriched for Villages 3–8. These new data are presented in Figures 5E, S11C, and S11D.

6. How to explain the clinical relevance of the identified tumor villages since multiple villages are observed in one patient?

Response: Thank you for this insightful question. As the reviewer noted, multiple villages can coexist within a single tumor, reflecting the well-documented intratumor heterogeneity within individual liver cancer patients. To assess the clinical relevance of this heterogeneity, we performed hierarchical clustering of tumors based on village-specific features. We found that despite the coexistence of multiple villages within individual tumors, patients could be stratified into two major groups with significantly different outcomes: one enriched for Villages 1 and 2 features and the other enriched for Villages 3–8 features. These findings suggest that while intratumor heterogeneity exists, certain dominant village-associated programs prevail at the patient level, allowing meaningful clinical stratification.

7. The tumor niche or microenvironment is a well-established concept to describe the tumor cell and surrounding immune cells, etc. A special paragraph is needed to discuss the similarities and differences between the tumor niche and tumor village concepts to help the readers understand this proposed new concept.

Response: We thank the reviewer for this thoughtful suggestion. The concept of the tumor microenvironment or niche is well established. The tumor microenvironment describes the broader ecosystem of non-malignant components surrounding malignant cells, while tumor niche refers to microanatomical structures that maintain specific tumor cells, such as stemness, dormancy, or metastatic potential. However, these concepts do not fully capture the coordinated spatial organization of diverse tumor cell states and their dynamic interactions with the microenvironment. To address this gap, we developed a graph attention neural network-based

method to identify spatially coordinated groups of tumor cell states, which we termed tumor cell “villages.” The village concept was inspired by human villages, which historically emerged to fulfill social, economic, and environmental needs, fostering community, adaptation, and resilience. Analogously, tumor cell villages focus specifically on malignant cells as the central actors, representing functional communities of diverse tumor cell states that are spatially organized and supported by dynamic microenvironments. Heterogeneous tumor cell states within a tumor cell village may act collectively to enhance survival, adaptation, and progression, thereby reducing the vulnerability of individual cells. We have added a paragraph in the Discussion section accordingly to further elaborate on these concepts and to help readers better understand this proposed new concept.

Minor comments:

1. For the reader's convenience, within the same figure (for example, Figure 2C), the same cell type should be marked with the same color. This will make it easier for the reader to match the cell's name with its position in the UMAP plot.

Response: We thank the reviewer for this helpful comment. We have reviewed all relevant figures and standardized the color schemes so that the same cell type is represented by the same color within each figure to improve readability.

2. Clustering of both tumor cells and non-tumor cells are not very clear to separate distinct cell types. There's a concern that batch effects in the spatial single-cell data may have been over-corrected during processing, potentially affecting the clustering result.

Response: Thank you for this comment. To address the concern of potential over-correction, we performed clustering of each non-malignant cell type for individual patients separately. We then compared these patient-derived clusters with the clusters obtained from Harmony integration. Overall, we found that patient-derived clusters predominantly mapped to single Harmony clusters, indicating that Harmony preserved biologically meaningful groupings. For malignant cells, clustering was performed based on module scores of distinct cellular state-related gene programs derived from the original dataset, without additional Harmony integration. This approach, which has been successfully applied in single-cell and spatial studies, allows us to identify shared transcriptional programs across patients (Barkley et al. Nat Genet 2022, PMID: 35931863; Greenwald et al. Cell 2024, PMID: 38653236). We have added the new analysis results to Figure S4.

3. In the abstract section, it is the tumor village, not the Spatial Dynamics Network, which is obtained through the deep learning-based strategy, as referred to the Methods sections.

Response: Thank you for this comment. We have modified it accordingly.

4. Typo: page 29, “recal” should be “recall”.

Response: Thank you for pointing out the typo. We have revised accordingly.

Reviewer #2:

1. One major improvement needed to ensure clarity and avoid misinterpretations:

Since the beginning, and already in the abstract, the authors speak about 50 samples. It is then going through the text, checking details in supplementary information, that readers then realize that in fact these 50 samples correspond to 7 patients. In the present version of the manuscript, this information is not evidently reported, as it appears only in Table S1. The number of analysed patients must be clearly stated since the beginning. Moreover, the authors stated that they performed studies on primary liver cancers. In fact, they used 3 iCCA and 4 HCC. iCCA and HCC have distinct molecular, histological, vulnerable features. Moreover, male and female HCC have often very distinct characteristics. Consequently, the study is based on a small fraction of patients. Considering the complexity of the study and the deep analyses performed, I do not find that 7 patients are not sufficient. The authors should clearly state these issues and clarify the samples on which the study has been performed, to avoid misunderstanding and misinterpretations from readers. I recommend as well to discuss these limitations in the discussion sections, in addition to a pertinent section in which the authors have already discussed some limitations of their study. I also want to stress that, despite the number and diversity of samples analysed could be a limitation of the study, the main findings and concept of 'villages' are supported by their existence across different primary liver cancer, and notably in relation the complexity and heterogeneity of liver cancer. Moreover, the authors wrote in the result section: “We noticed that tumor cell village compositions were much more similar across different sampling regions from the same patient compared to those from different patients, suggesting relatively stable tumor cell populations within individual patients”. As I mentioned above, this may be again related to the comparison of patients with distinct primary liver cancer types (HCC and iCCA) and from male versus female patients. The authors should widen their interpretation, as the one currently proposed would be only supported by analysing several patients belonging to the same primary liver cancer type and sex of patients. This additional aspect could be as well elaborated in the discussion section. To help readers interpreting differences identified in studies, I recommend to reports in all panels in figures and supplementary figures, which samples belong to HCC and to iCCA patients.

Response: We sincerely appreciate the reviewer's thoughtful comments and valuable suggestions. We also thank the reviewer for the positive assessment of our tumor cell village concept and the data analyses supporting it. We fully agree that clarity regarding the number and types of patients analyzed is critical to avoid misinterpretation. Accordingly, we have revised the figures and main texts to clearly indicate the number of HCC and iCCA patients included in this study. In addition,

we specify the clinical subtype of each sample in the main text and all figure panels, with sample IDs labeled with "H" for HCC and "C" for iCCA. We also acknowledge the limitation of our relatively small discovery cohort, as well as the potential influence of factors such as patient sex, etiology, and primary cancer type on tumor heterogeneity. Tumors from different patients evolve independently, leading to unique gene expression profiles shaped by patient-specific characteristics. This heterogeneity between patients may reflect broader molecular, histological, etiological, and demographic complexity, in addition to individual variations. These considerations further underscore the value of our tumor cell village concept. By spatially characterizing the landscapes of individual tumors, the village concept enables a deeper understanding of how tumor cells establish unique landscapes within individual patients, while also facilitating comparisons of these landscapes across patients. Moreover, defining tumor cell villages may uncover unique molecular co-dependencies among cells that underlie the spatial organization of each village, potentially revealing new opportunities for therapeutic intervention. As the reviewer suggested, we have expanded the Discussion section to more explicitly address the limitations of our study and we further emphasize the importance of understanding tumor spatial landscape through defining tumor cell villages. Please refer to the revised Figure 1A and the revised text in the Introduction, Results, and Discussion sections.

Minor comments:

1. Genes should be in *Italic*.

Response: Thank you for this comment. We have revised gene names in figures and the manuscript accordingly, using *italic* for gene names and regular font for proteins.

2. in S1, "B" should be positioned higher to make evident for readers that "fraction of cells" and "average expression" scales are referred to graphs below.

Response: We have modified this figure accordingly. We thank the reviewer again for all the helpful suggestions.

Reviewer #3:

Major comments:

1. In Figure 1A, the authors described multi-point tumor tissue sampling from the cancer region, boundary zone, and adjacent normal area for each patient. However, the paper provides minimal discussion on the differences among these three regions and the spatial variations within the cancer region itself. An in-depth analysis of these differences might better reveal intratumoral heterogeneity and address actual biological and clinical problems. This would be more meaningful than merely optimizing existing computational methods or showcasing fancier computational approaches.

Response: We appreciate the reviewer for this suggestion. Accordingly, we have performed an in-depth analysis to investigate spatial variations both among different tumor regions and within individual regions. Specifically, we first conducted a detailed comparison of cellular compositions across different tumor regions by examining the major cell types, malignant cell states, and non-malignant cell subtypes. At the level of major cell types, we observed that T cells were markedly decreased in the tumor border and core relative to the adjacent non-tumor region, whereas myeloid cells and fibroblasts progressively increased from the border to the core. Epithelial cells also showed a gradual decrease in the border and core compared with the adjacent non-tumor region. Detailed comparison of tumor cell states revealed largely comparable proportions between the border and core regions in our cohort. While most non-malignant cell subtypes showed no significant differences across these regions, several displayed region-specific enrichment. At the tumor border, two immune subtypes (Kupffer cells and $CD8^+$ effector T cells) and four stromal subtypes (liver sinusoidal endothelial cells [LSECs], post-capillary venule endothelial cells, $CCL19^+$ CAFs, and $CXCL12^+$ CAFs) were significantly enriched compared to the tumor core. This observation is consistent with the typical localization of Kupffer cells and LSECs in normal liver tissue. The additional enrichment of $CD8^+$ effector T cells, inflammatory CAFs ($CCL9^+$ CAFs and $CXCL12^+$ CAFs), and post-capillary venule endothelial cells highlights an inflamed microenvironment in this region. In contrast, tip cells and smooth muscle cells were elevated in the tumor core, along with immune-regulatory TAMs (Reg-TAMs), suggesting a potential pro-angiogenic and immunosuppressive microenvironment.

Additionally, we investigated the spatial variation of cells within each tumor region by calculating Moran's I to quantify spatial autocorrelation. For all major cell types, Moran's I values were positive, indicating that each cell type exhibits some degree of spatial autocorrelation within individual tumor regions. Notably, Moran's I increased for epithelial cells, myeloid cells, fibroblasts, and endothelial cells in the tumor border and core compared with adjacent non-tumor regions, suggesting stronger spatial clustering of these populations within tumors. In contrast, Moran's I values for T cells and B cells decreased in the tumor border and core relative to non-tumor regions, reflecting a more dispersed distribution within the tumor. These observations are consistent with the clustered distribution of T cells and B cells around portal veins in normal liver, whereas other cell types are more evenly dispersed in normal tissue. Malignant cell states displayed low but positive Moran's I values, with no significant differences between border and core, suggesting only marginal spatial autocorrelation of each malignant cell state within individual tumor regions for the samples in our cohort. These new data have been shown in Figures S6 and S7 and we have also revised the texts in the main manuscript accordingly.

2. The tumor cell states obtained through NMF in Figure 2A contain multiple stress states. The first question is, given that Stress/Metal and Stress are adjacent, why separate them instead of combining them into a single Stress state? The second question is, the stress comprised of four

clusters shows results in Figure S2E that are mainly concentrated in the 2CB samples. Is this a sample-specific phenomenon? Including such information may not be conducive to explaining universal phenomena. The authors should filter out patient- or sample-specific states before proceeding with subsequent analysis.

Response: Thank you for this comment. When determining malignant cell state, we calculated module scores for each tumor cluster and further annotated each cluster based on the dominant modules with the highest scores. As shown by the log₂ fold change of Stress and Metal module scores below, Cluster 10 had comparable enrichment for both modules, whereas Clusters 16, 21, 24, and 7 showed much higher Stress scores than Metal scores. Therefore, we annotated Cluster 10 as Stress/Metal and others as Stress.

In Figure S2E, the Stress state appears as a dominant malignant cell state only in sample 2CB. This bar plot reflects the relative proportions of different cellular states within each sample. While it allows us to compare overall cellular compositions, it does not reflect the actual number of cells across samples. To address this limitation, we have now provided a table reporting the absolute number of malignant cells in each state across individual samples. As shown in this table, Stress state cells are present across different samples and their absolute numbers are relatively large, suggesting that Stress state cells are not sample-specific. The data are presented in Table S4.

3. Can the numerous cell subtypes identified in Figure 2C be mapped to the cell subtypes currently recognized in the pan-cancer cell atlas?

Response: Thank you for raising this important question. We annotated cell subtypes within each major non-malignant cell type based on differentially expressed genes and further used publicly available cell atlases as references. Accordingly, the identified cell subtypes are well mapped to publicly available pan-cancer cell atlases. Specifically, for endothelial cell (EC) subtypes, tip cells (*CXCR4*/collagen genes), arterial ECs (*EFNB2*), and lymphatic ECs (*PROX1*) have been identified

in across multiple cancer types (Li et al. Natl Sci Rev 2024, PMID: 39345334; Goveia et al. Cancer Cell 2020, PMID: 31935371; Vanlandewijck et al. Nat 2018, PMID: 29443965; Zhao et al. Nat Cancer 2024, PMID: 38172338). Postcapillary veins (*VCAMI/CLU*), activated capillaries (*CAVI/IGFBP7*), *CD14*⁺ capillaries (*CD14*), and scavenging capillaries (*CD68*) have been described in lung cancer (Goveia et al. Cancer Cell 2020, PMID: 31935371). LSEC (*LYVE1*) are unique to the liver (MacParland et al. Nat Commun 2018, PMID: 30348985). We also identified a cluster of endothelial cells showing high expression of *RGCC* but lacking well-established markers, and therefore we annotated the cluster as unknown. We identified eight subtypes within the fibroblast population. Among them, mCAFs (collagen genes), apCAFs (*CD74* and antigen-presentation), and three inflammatory subtypes (*CCL9*⁺ CAFs, *IL32*⁺ CAFs, and *CXCL12*⁺ CAFs) expressing high level of inflammatory and chemo-attractive mediators have been described in various cancer types (Liu et al. Cancer Cell 2025, PMID: 40154487; Cords et al. Nat Commun 2023, PMID: 37463917; Zhang et al. J Hepatol 2020, PMID: 32505533; Lavie et al. Nat Cancer 2022, PMID: 35883004). eCAFs (*KRT17*) has been reported in iCCA (Zhang et al. J Hepatol 2020, PMID: 32505533), while tCAFs (*VEGFA*) have been identified in breast and lung cancers (Cords et al. Nat Commun 2023, PMID: 37463917; Cords et al. Cancer Cell 2024, PMID: 38242124). In addition, smooth muscle cells (*RGS5/ACTA2*) are commonly observed in various normal and cancer tissues. Among the ten myeloid cell subtypes, nine subtypes including Prolif-TAM (*HMG2/TUBB*), Reg-TAM (*GPNMB*), Angio-TAM (*TIMP1*), INT monocyte (*SERPINA1*), LA-TAM (*APOE*), TIM (*CD14*), and Inflamm-TAM (*CCL20*), neutrophils (*CSF3R*) and mast cells (*CPA3*) have been reported in various cancer studies (Cheng et al. Cell 2021, PMID: 33545035, Xue et al. Nat 2022, PMID: 36352227, Ma et al. Trends Immunol 2022, PMID: 35690521). Kupffer cells (*CD5L*) are unique to the liver (Aizarani et al. Nat 2019, PMID: 31292543). For the T cell subtypes, T_{eff} (*GZMK/GZMB/GNLY*), T_{mem} (*ANXA1*), T_{reg} (*FOXP3*), T_{EMRA} (*IL7R/GZMK*), Prolif-T (*MKI67*) and IFN-I responsive T (*IFIT1*) were annotated based on gene markers obtained from a pan-cancer study of T cells across 21 cancer types as well as T cell studies of individual cancer types (Zheng et al. Science 2021, PMID: 34914499; Zheng et al. Cell 2017, PMID: 28622514; Ma et al. J Hepatol 2021, PMID: 34216724). *LGALS1*⁺, *EOMES*⁺, *CXCR6*⁺, and *JUN*⁺ were annotated based on their highly expressed genes. We also observed three clusters of T cells lacking classical markers and we therefore annotated them as unknown. The five B cell subtypes were annotated based on the markers from pan-cancer studies of B cells (Yang et al. Cell 2024, PMID: 39047727; Ma et al. Science 2024 PMID: 38696569).

4. As shown in Figure 5C, the authors ultimately identified 8 tumor villages. In Figure S7B, the authors demonstrated the tumor village composition of each sample, showing that V1 is a village specific to patient 2H, while V2 is specific to patient 3H. Looking at the SDNs in Figure 5C, T3, 4, and 5 are specific to V1, while T1 and 2 are specific to V2. Since I couldn't find the composition of T across samples, I can only speculate whether V1 (T3, 4, 5) and V2 (T1, 2) respectively represent spatial patterns specific to patients 2H and 3H? If this is indeed the case, it seems

unreasonable to combine these two patient-specific tumor villages into one group for prognostic analysis.

Response: We thank the reviewer for this comment. We agree with the reviewer that patient 2H is dominated by Village 1 and patient 3H by Village 2. However, we also observed that Villages 1 and 2 were present at high proportions in other patients (Village 1 in patient 2C and Village 2 in patient 1H). We recognized that the relatively small number of patients in our discovery cohort is a limitation, which may give the appearance that certain villages are restricted to only a few patients. To address this concern, we extended our analysis by applying Villages 1 and 2 signatures to independent datasets, including 10x Visium spatial transcriptome data from 46 samples across four liver cancer studies (three HCC studies and one iCCA study), scRNA-seq data from 37 liver cancer patients, and bulk transcriptome data from 674 liver cancer patients. Across these larger datasets, Villages 1 and 2 signatures were frequently observed across different patients, demonstrating that these tumor cell villages are not specific to a few patients. Additionally, their consistent association with clinical outcomes across different patient cohorts further supports that Villages 1 and 2 are biologically meaningful. The data are presented in Figures 5E-5H and S11A-S11E. We have revised the manuscript accordingly and also expanded the Discussion section to reflect the limitations of our study.

5. Furthermore, according to the description, both V1 and V2 belong to a vascularized environment. Theoretically, an environment associated with vascularization should be related to tumor metastasis and consequently linked to poor prognosis. However, the authors' research results show that this is associated with good prognosis - how can this phenomenon be explained?

Response: We appreciate the reviewer's comment and agree with the reviewer that tumor angiogenesis is usually linked to poor patient outcomes. To investigate this question, we further examined tumor vasculature in the identified tumor cell villages. We found that although Villages 1 and 2 showed overall enrichment in endothelial cells, the expression of tumor-specific endothelial signature genes (derived from Croix et al. Science 2000, PMID: 10947988) was significantly higher in Villages 3–8 compared with Villages 1 and 2. Previous single-cell studies have demonstrated that endothelial cells in cancer are heterogeneous, with some subtypes resembling normal vasculature and others displaying tumor-promoting properties (Li et al. Natl Sci Rev 2024, PMID: 39345334). Our results suggest that the endothelial cells in Villages 3–8 may acquire more tumor-associated features, which may underlie the poorer prognosis observed in these patients. The new data are presented in Figure S11G.

6. The author introduces the concept of molecular co-dependencies at the end of the article, identifying multiple gene pairs through correlation analysis and suggesting these gene pairs are factors that maintain village stability. However, I notice that these gene pairs are not ligand-receptor pairs. If the genes within these pairs cannot directly interact with each other, then the

claim that such pairing can stabilize the village seems rather far-fetched. This correlation is more likely due to spatial proximity, leading to related phenotypes in their respective cell types. Therefore, rather than attributing it to the action of gene pairs, it would be more accurate to say that the coexistence of two cell types facilitates the formation of villages. I believe that according to the concept proposed by the author, the gene pairs identified through analysis need to be explained through molecular and cellular mechanisms, which currently seem weak and require more direct evidence.

Response: We greatly appreciate the reviewer's insightful comment. Our initial motivation was to understand molecular co-dependencies within individual tumor cell villages that extend beyond ligand–receptor interactions. While ligand–receptor signaling captures an important subset of intercellular communication, cells also interact through additional mechanisms such as extracellular vesicles, metabolites, adhesion molecules, gap junctions, and nanotubes. By examining molecular co-dependencies, our goal was to capture broader molecular relationships that may underlie village stability. We fully agree with the reviewer that the gene pairs we identified were based on spatial correlation and that elucidating the underlying molecular and cellular mechanisms is critical for understanding the biology of tumor cell villages. In the original manuscript, we highlighted *SPINK1* in tumor cells and its associated genes in mCAFs as a proof of concept. We further demonstrated that *SPINK1*+ tumor cells engaged more extensively with mCAFs than with other CAF subtypes through ligand-receptor interactions. We provided only the top 20 ligand–receptor interactions between *SPINK1*+ tumor cells and CAFs in the manuscript, leaving many additional interactions unexplored. In response to the reviewer's comment, we now provide all the identified ligand–receptor interactions. Notably, *SPINK1* itself was identified as one of the ligands in these interactions with *EGFR* serving as its receptor. Consistently, *SPINK1* downstream genes were expressed at significantly higher levels in mCAFs compared with other CAFs. Similarly, the *EGFR* downstream genes were significantly upregulated in mCAFs. We also performed immunofluorescence staining for *SPINK1* and *COL1A1* (a representative marker for mCAF) to confirm the colocalization of the two proteins. Previous studies have demonstrated that *SPINK1* can stimulate the proliferation of both tumor cells and fibroblasts through binding to *EGFR* and activating its downstream signaling using experimental approaches (Ateeq et al. *Sci Transl Med* 2011, PMID: 21368222; Ozaki et al, *Mol Cancer Res* 2009, PMID: 19737965). It is therefore likely that *SPINK1* may also promote the proliferation of mCAFs via *EGFR* signaling, providing molecular mechanisms for the *SPINK1*-related gene pairs identified in our study. The new results have been added to Figure S12 and Tables S7 and S8. As we only demonstrated the molecular mechanisms of *SPINK1* as a proof of concept in this study, we have expanded the Discussion to acknowledge the limitation of our study and to emphasize the need for further experimental approaches to elucidate the molecular and cellular mechanisms of the identified gene pairs. We thank the reviewer again for this valuable comment, which is very helpful to improve our manuscript.

Minor comments:

1. Please label the sample size information from Table S1 in Figure 1A. While the article studies intratumor heterogeneity, the schematic diagram creates confusion by showing only one sample each from tumor, boundary, and normal tissue.

Response: Thank you for pointing this out. We have revised Figure 1A accordingly.

2. While I understand the author's intention to make the visualization more prominent, the scores in the six graphs in Figure 1D have been adjusted too artificially and no longer appear to be continuous values.

Response: We appreciate the comment and have modified the color in Figure 1D to continuous values accordingly.

3. The authors present multiple clustering results in the paper, all simply named c1, c2, etc. Although Figure S4 indicates the subtype classification of each cell type, Figure 2A does not clearly specify that these are tumor subtypes. It is suggested to add abbreviations of the major categories before c to avoid confusion between different clustering results.

Response: Thank you for this comment. We have modified the cluster naming scheme in Figure 2A, assigning the prefix “tc” to tumor clusters for clarity. We thank the reviewer again for all the constructive and helpful comments to improve our paper.

Responses to Reviewers' Comments

Reviewer #1:

All my concerns have been addressed. I have no further comments.

Reviewer #2:

The authors have adequately addressed the points raised. Congratulation for this inspiring work.

Response: We thank the reviewers again for all the constructive and helpful comments to improve our paper.

Reviewer #3:

The authors have basically answered all the questions. However, I still have a minor question about the response to the first comment. Regarding the epithelial cells in the Supplementary Figure 6A, according to the description in line 579 of the manuscript, these cells include hepatocytes, cholangiocytes, and tumor cells. Among them, normal epithelial cells should be enriched in the adjacent non-tumor area, while tumor cells should be enriched in the tumor area. Therefore, observing the regional enrichment pattern of total epithelial cells, which include both normal and cancerous epithelial cells, seems unreasonable.

Response: Thank you for this comment. We initially presented the total epithelial abundance in Supplementary Figure 6A to provide an overview of epithelial cell changes across different tumor regions. In response to the reviewer's comment, we have now added the detailed abundances of malignant and non-malignant (hepatocytes and cholangiocytes) epithelial cells. As expected, non-malignant epithelial cells were predominantly located in the adjacent non-tumor region, whereas malignant cells were mainly enriched in the tumor core. Please refer to the revised Supplementary Figure 6A. We thank the reviewer again for this helpful suggestion.

Responses to Reviewers' Comments

Reviewer #3:

I have no further comments.

Response: We thank the reviewer again for all the helpful comments to improve our paper.